# A Novel Framework for PAC-Bayes Derandomization: Applications to Majority Votes

## Abstract

PAC-Bayes is a popular and efficient framework for obtaining generalization guarantees in situations involving uncountable hypothesis spaces. Unfortunately, in its classical formulation, it only provides guarantees on the expected risk of a randomly sampled hypothesis. This requires stochastic predictions at test time, making PAC-Bayes unusable in many practical situations where a single deterministic hypothesis must be deployed. We propose a unified framework to extract guarantees holding for a single hypothesis from PAC-Bayesian guarantees. We present a general oracle bound, and derive from it a numerical bound and a specialization to majority vote. We empirically show that our approach consistently outperforms popular baselines (by up to a factor of 2) when it comes to generalization bounds for single classifiers.

## 1 Introduction

The PAC-Bayes theory, initiated by McAllester (1998; 2003) and enriched by many (see Alquier (2024) for a recent survey), has become a prominent framework for obtaining non-vacuous generalization guarantees on uncountable hypothesis spaces. Such hypothesis spaces include linear classifiers (Langford & Shawe-Taylor, 2002; Germain et al., 2009), weighted majority vote (Roy et al., 2011; Bellet et al., 2014; Zantedeschi et al., 2021) and neural networks (Dziugaite & Roy, 2017; Letarte et al., 2019; Pérez-Ortiz et al., 2021a;b; Leblanc et al., 2025).

In contrast to classical PAC bounds (Valiant, 1984), such as those based on the concept of VC-dimension (Vapnik, 2000) or Rademacher complexity (Mohri et al., 2012), PAC-Bayes bounds involve prior and posterior distributions over the hypothesis space, the latter usually being learned by minimizing a generalization guarantee as a training objective (Germain et al., 2009; Parrado-Hernández et al., 2012; Wu et al., 2021). Classical PAC-Bayesian guarantees do not hold for every single hypothesis simultaneously or for a single one, but bound the expected risk of a randomly drawn hypothesis with respect to the posterior distribution. This provides aggregate information about the hypothesis space weighted by the learned posterior distribution. In many applications, a single hypothesis must be used, for which classical PAC-Bayes cannot provide generalization guarantees. For instance, in breast cancer diagnosis (Naji et al., 2021), a patient must receive a single, consistent diagnosis; randomly varying predictions would erode trust and violate medical protocols. Similarly, in situations where interpretability is of importance, say for fairness concerns (Dastin, 2018; Telford, 2019; Martinez & Kirchner, 2021), stochasticity could obfuscate the prediction mechanism.

On the other hand, most classical approaches to generalization bounds stand for all hypotheses simultaneously, thus are either expressed as a worst-case analysis (Vapnik, 2000; Mohri et al., 2012) or union bounds (a classical application lies in the extension of Hoeffding's inequality over a countable set of hypotheses (Shalev-Shwartz & Ben-David, 2014); many results in the sample compression literature are inherently clever union bounds, e.g., Marchand & Shawe-Taylor (2002); Bazinet et al. (2025); etc.) This is unnecessarily conservative, since ultimately, we only wish for the risk of a single carefully chosen hypothesis to be bounded, not all of them.

**Contributions.** We propose a unified framework for obtaining generalization bounds on the risk of single hypotheses using PAC-Bayes bounds on a weighted hypothesis space as a building block. This conversion, which we call *Derandomization*, leverages the tightness of PAC-Bayes bounds to address concerns about their practical use. This is achieved by first deriving an oracle bound that is independent of the hypothesis space and the form of the prior and posterior distributions. Then, we specialize this bound to the case of majority votes, using Categorical, Dirichlet, or Gaussian distributions. Finally, we evaluate (and compare to several baselines) our approach on many binary and multi-class classification tasks, providing empirical evidence of its tightness.

## 2 Background and definitions

**Prediction problem.** A dataset $S = \{(\mathbf{x}_j, \mathbf{y}_j)\}_{j=1}^m$ consists of $m$ examples, each being a feature-label pair $(\mathbf{x}, \mathbf{y}) \in \mathcal{X} \times \mathcal{Y}$. We focus on (possibly multivariate) classification tasks, denote $k$ the number of classes, and let $\mathcal{Y}$ be the set of the $k$ different one-hot vectors of length $k$. A predictor is a function $\mathbf{h} : \mathcal{X} \to \mathcal{Y}'$, from a predictor space $\mathcal{H}$, with $\mathcal{Y}' \supseteq \mathcal{Y}$. We consider a binary loss function $\ell : \mathcal{Y}' \times \mathcal{Y} \to \{0, 1\}$. We denote $\mathcal{D}$ the data-generating distribution over $\mathcal{X} \times \mathcal{Y}$ such that $S \sim \mathcal{D}^m$.

Given a predictor $\mathbf{h} \in \mathcal{H}$ and a loss function $\ell$, the empirical loss of the predictor over a set of $m$ *i.i.d.* examples is

$$R_S(\mathbf{h}) = \frac{1}{m} \sum_{(\mathbf{x}, \mathbf{y}) \in S} \ell(\mathbf{h}(\mathbf{x}), \mathbf{y}),$$

while the generalization loss (*risk*) of a predictor $\mathbf{h}$ is

$$R_\mathcal{D}(\mathbf{h}) = \mathop{\mathbb{E}}_{(\mathbf{x}, \mathbf{y}) \sim \mathcal{D}} [\ell(\mathbf{h}(\mathbf{x}), \mathbf{y})].$$

### 2.1 PAC-Bayesian learning framework

A defining characteristic of PAC-Bayes bounds is that they rely on prior $P$ and posterior $Q$ distributions over the predictor space $\mathcal{H}$. Hence, most PAC-Bayes results are expressed as upper bounds on the $Q$-expected loss of the predictor space ($R_\mathcal{D}(Q) = \mathbb{E}_{\mathbf{h} \sim Q} R_\mathcal{D}(\mathbf{h})$, referred to as the *Gibbs risk*). A typical PAC-Bayes bound states that for any prior $P$, with probability at least $1 - \delta$, for all posteriors $Q$ over $\mathcal{H}$:

$$R_\mathcal{D}(Q) \leq f\Big(R_S(Q), \mathrm{KL}(Q||P), m, \delta\Big),$$

for some function $f$, where $R_S(Q) = \mathop{\mathbb{E}}_{\mathbf{h} \sim Q} R_S(\mathbf{h})$.

## 3 Derandomization bounds

We formalize the capacity to bound the risk of a single classifier with the help of a bound on the Gibbs risk by introducing *Derandomization* bounds.

**Definition** (Derandomization bound). For some hypothesis space $\mathcal{H}$, distribution $Q$ over $\mathcal{H}$, and hypothesis $\mathbf{h} \in \mathcal{H}$, a Derandomization bound is an upper bound on $R_\mathcal{D}(\mathbf{h})$ which depends on $R_\mathcal{D}(Q)$ and takes the following general form, for some function $f$:

$$R_\mathcal{D}(\mathbf{h}) \leq f\Big(R_\mathcal{D}(Q)\Big).$$

### 3.1 A general bound

We first provide a general relationship between the risk of a single classifier and the Gibbs risk. To do so, we first decompose the Gibbs risk into two terms by conditioning on whether a given hypothesis $\mathbf{h}$ makes

an error:

$$\mathfrak{o}_{\mathcal{D}}^Q(\mathbf{h}) = \mathop{\mathbb{E}}_{(\mathbf{x},\mathbf{y})\sim\mathcal{D}} \left[ \mathop{\mathbb{E}}_{\mathbf{h}'\sim Q}\ell(\mathbf{h}'(\mathbf{x}),\mathbf{y}) \;\middle|\; \ell(\mathbf{h}(\mathbf{x}),\mathbf{y})=0 \right],$$

$$\mathfrak{i}_{\mathcal{D}}^Q(\mathbf{h}) = \mathop{\mathbb{E}}_{(\mathbf{x},\mathbf{y})\sim\mathcal{D}} \left[ \mathop{\mathbb{E}}_{\mathbf{h}'\sim Q}\ell(\mathbf{h}'(\mathbf{x}),\mathbf{y}) \;\middle|\; \ell(\mathbf{h}(\mathbf{x}),\mathbf{y})=1 \right],$$

where $\mathfrak{o}_{\mathcal{D}}^Q(\mathbf{h})$ is the Gibbs risk on $\mathbf{h}$'s correct predictions, and $\mathfrak{i}_{\mathcal{D}}^Q(\mathbf{h})$ is the Gibbs risk on $\mathbf{h}$'s incorrect predictions.[1] Intuitively, if $\mathbb{E}_{\mathbf{h}'\sim Q}\ell(\mathbf{h}'(\mathbf{x}),\mathbf{y})$ is positively correlated with $\ell(\mathbf{h}(\mathbf{x}),\mathbf{y})$, then we should have that $\mathfrak{o}_{\mathcal{D}}^Q(\mathbf{h}) \le \mathfrak{i}_{\mathcal{D}}^Q(\mathbf{h})$.

**Proposition 1** (Stochastic-deterministic relation)**.** *For any data distribution $\mathcal{D}$, distribution $Q$ over $\mathcal{H}$ and classifier $\mathbf{h} \in \mathcal{H}$ such that $\mathfrak{i}_{\mathcal{D}}^Q(\mathbf{h}) \neq \mathfrak{o}_{\mathcal{D}}^Q(\mathbf{h})$, we have*

$$R_{\mathcal{D}}(\mathbf{h}) = \frac{R_{\mathcal{D}}(Q) - \mathfrak{o}_{\mathcal{D}}^Q(\mathbf{h})}{\mathfrak{i}_{\mathcal{D}}^Q(\mathbf{h}) - \mathfrak{o}_{\mathcal{D}}^Q(\mathbf{h})}.$$

See Section B for complete demonstrations of all mathematical results presented in the article.

*Proof sketch.* We have $R_{\mathcal{D}}(Q) = \mathfrak{i}_{\mathcal{D}}^Q(\mathbf{h}) \cdot R_{\mathcal{D}}(\mathbf{h}) + \mathfrak{o}_{\mathcal{D}}^Q(\mathbf{h}) \cdot (1 - R_{\mathcal{D}}(\mathbf{h}))$. Solving for $R_{\mathcal{D}}(\mathbf{h})$ leads to the main result. $\square$

Classical PAC-Bayesian tools bound $R_{\mathcal{D}}(Q)$ with respect to the distribution $\mathcal{D}$. However, $\mathfrak{o}_{\mathcal{D}}^Q(\mathbf{h})$ and $\mathfrak{i}_{\mathcal{D}}^Q(\mathbf{h})$ are related to conditional distributions $\mathcal{D}|\{\ell(\mathbf{h}(\mathbf{x}),\mathbf{y})=0\}$ and $\mathcal{D}|\{\ell(\mathbf{h}(\mathbf{x}),\mathbf{y})=1\}$, so existing PAC-Bayesian bounds cannot be used directly for bounding those quantities. Theorem 2 introduces a modification of the well-known Seeger's bound (Seeger, 2002) (see Theorem 13 in appendix) that bounds both $\mathfrak{o}_{\mathcal{D}}^Q(\mathbf{h})$ and $\mathfrak{i}_{\mathcal{D}}^Q(\mathbf{h})$.

**Theorem 2** (Conditional PAC-Bayes)**.** *Let $\mathrm{kl}(q,p) = q\ln\left(\frac{q}{p}\right) + (1-q)\ln\left(\frac{1-q}{1-p}\right)$. For any data distribution $\mathcal{D}$, hypothesis $\mathbf{h} \in \mathcal{H}$, prior distribution $\mathcal{P}$ over $\mathcal{H}$, and $\delta \in (0,1]$, both the following statements hold with probability at least $1-\delta$ over the draw $S \sim \mathcal{D}^m$, for all $Q$ over $\mathcal{H}$:*

$$\mathrm{kl}\left(\mathfrak{o}_S^Q(\mathbf{h}) \,\middle\|\, \mathfrak{o}_{\mathcal{D}}^Q(\mathbf{h})\right) \le \frac{1}{m_{(\mathbf{h},0)}}\left[ \mathrm{KL}(Q\|P) + \ln\left(\frac{2\sqrt{m}}{\delta}\right)\right],$$

$$\mathrm{kl}\left(\mathfrak{i}_S^Q(\mathbf{h}) \,\middle\|\, \mathfrak{i}_{\mathcal{D}}^Q(\mathbf{h})\right) \le \frac{1}{m_{(\mathbf{h},1)}}\left[ \mathrm{KL}(Q\|P) + \ln\left(\frac{2\sqrt{m}}{\delta}\right)\right],$$

*where $\mathfrak{o}_S^Q(\mathbf{h})=R_{S_{(\mathbf{h},0)}}(Q)$, $\mathfrak{i}_S^Q(\mathbf{h})=R_{S_{(\mathbf{h},1)}}(Q)$, $S_{(\mathbf{h},\cdot)}=\{(\mathbf{x},\mathbf{y})\in S : \ell(\mathbf{h}(\mathbf{x}),\mathbf{y})=\cdot\}$, and $|S_{(\mathbf{h},\cdot)}|=m_{(\mathbf{h},\cdot)}$.*

The following result uses Theorem 2 to bound the quantities involved in Proposition 1; it bounds the deterministic risk of any hypothesis using three PAC-Bayesian bounds simultaneously.

**Corollary 3** (Triple bound – Single hypothesis)**.** *Let $\mathcal{D}$ be a data distribution, $Q$ a distribution over $\mathcal{H}$, $\mathbf{h} \in \mathcal{H}$ and $\delta_1, \delta_2, \delta_3 \in [0,1]$. Let $\widetilde{R}_S(Q)$, $\widetilde{\mathfrak{o}}_S^Q(\mathbf{h})$ and $\widetilde{\mathfrak{i}}_S^Q(\mathbf{h})$ be, respectively, probabilistic upper, lower, and lower bounds on $R_{\mathcal{D}}(Q)$, $\mathfrak{o}_{\mathcal{D}}^Q(\mathbf{h})$, and $\mathfrak{i}_{\mathcal{D}}^Q(\mathbf{h})$ such that*

$$(1) \quad \mathop{\mathbb{P}}_{S\sim\mathcal{D}^m}\left(\forall Q \text{ over } \mathcal{H} : R_{\mathcal{D}}(Q) \le \widetilde{R}_S(Q)\right) \ge 1 - \delta_1\,,$$

$$(2) \quad \mathop{\mathbb{P}}_{S\sim\mathcal{D}^m}\left(\forall Q \text{ over } \mathcal{H} \;:\; \mathfrak{o}_{\mathcal{D}}^Q(\mathbf{h}) \ge \widetilde{\mathfrak{o}}_S^Q(\mathbf{h})\right) \ge 1 - \delta_2\,,$$

$$(3) \quad \mathop{\mathbb{P}}_{S\sim\mathcal{D}^m}\left(\forall Q \text{ over } \mathcal{H} \;:\; \mathfrak{i}_{\mathcal{D}}^Q(\mathbf{h}) \ge \widetilde{\mathfrak{i}}_S^Q(\mathbf{h})\right) \ge 1 - \delta_3\,,$$

$$(4) \quad \widetilde{\mathfrak{i}}_S^Q(\mathbf{h}) > \widetilde{\mathfrak{o}}_S^Q(\mathbf{h})\,.$$

---

[1]We discuss how to generalize these quantities to any (possibly unbounded) loss in Section 8.

*Then,*

$$\mathop{\mathbb{P}}_{S\sim\mathcal{D}^m}\left(\forall Q \ over \ \mathcal{H} \ : \ R_{\mathcal{D}}(\mathbf{h}) \leq \frac{\widetilde{R}_S(Q) - \widetilde{\mathfrak{o}}_S^Q(\mathbf{h})}{\widetilde{\mathfrak{i}}_S^Q(\mathbf{h}) - \widetilde{\mathfrak{o}}_S^Q(\mathbf{h})}\right) \geq 1 - \delta \,,$$

*where $\delta = \delta_1 + \delta_2 + \delta_3$.*

Note that Corollary 3 only holds for a hypothesis $\mathbf{h}$ chosen before seeing $S$. Thus, it cannot be used to select $\mathbf{h}$ based on $S$, as the probabilistic bounds on $\mathfrak{o}_{\mathcal{D}}^Q(\mathbf{h})$ and $\mathfrak{i}_{\mathcal{D}}^Q(\mathbf{h})$ would not be valid. Corollary 4 addresses this by replacing probabilistic lower bounds on $\mathfrak{o}_{\mathcal{D}}^Q(\mathbf{h})$ and $\mathfrak{i}_{\mathcal{D}}^Q(\mathbf{h})$ with deterministic lower bounds that hold for all $\mathbf{h}$ simultaneously. This enables data-dependent hypothesis selection (Section 4).

**Corollary 4** (Triple bound)**.** *Let $\mathcal{D}$ be a data distribution, $Q$ a distribution over $\mathcal{H}$ and $\delta \in [0,1]$. Let $\widetilde{R}_S(Q)$, $\widetilde{\mathfrak{o}}_S^Q(\mathbf{h})$ and $\widetilde{\mathfrak{i}}_S^Q(\mathbf{h})$ be such that*

$$(1) \quad \mathop{\mathbb{P}}_{S\sim\mathcal{D}^m}\left(\forall Q \ over \ \mathcal{H} \ : \ R_{\mathcal{D}}(Q) \leq \widetilde{R}_S(Q)\right) \geq 1 - \delta \,,$$

$$(2) \quad \forall Q \ over \ \mathcal{H}, \mathbf{h} \in \mathcal{H} \ : \ \mathfrak{o}_{\mathcal{D}}^Q(\mathbf{h}) \geq \widetilde{\mathfrak{o}}_S^Q(\mathbf{h}) \ and \ \mathfrak{i}_{\mathcal{D}}^Q(\mathbf{h}) \geq \widetilde{\mathfrak{i}}_S^Q(\mathbf{h}) \,,$$

$$(3) \quad \widetilde{\mathfrak{i}}_S^Q(\mathbf{h}) > \widetilde{\mathfrak{o}}_S^Q(\mathbf{h}) \,.$$

*Then,*

$$\mathop{\mathbb{P}}_{S\sim\mathcal{D}^m}\left(\forall Q \ over \ \mathcal{H}, \mathbf{h} \in \mathcal{H} \ : \ R_{\mathcal{D}}(\mathbf{h}) \leq \frac{\widetilde{R}_S(Q) - \widetilde{\mathfrak{o}}_S^Q(\mathbf{h})}{\widetilde{\mathfrak{i}}_S^Q(\mathbf{h}) - \widetilde{\mathfrak{o}}_S^Q(\mathbf{h})}\right) \geq 1 - \delta \,.$$

In the following section, we restrict ourselves to given hypothesis sets $\mathcal{H}$ that enable direct computation of $\widetilde{R}_S(Q)$, $\widetilde{\mathfrak{o}}_S^Q(\mathbf{h})$ and $\widetilde{\mathfrak{i}}_S^Q(\mathbf{h})$, so that Corollary 4 can be used as a training objective.

## 4 Derandomization bounds for stochastic majority vote

Weighted majority vote is a central technique for combining predictions of multiple classifiers, for example in random forests (Breiman, 1996; 2001), boosting (Freund & Schapire, 1996), gradient boosting (Mason et al., 1999; Friedman, 2002), and when combining predictions of heterogeneous classifiers. It is part of the winning strategies in many machine learning competitions. The power of the majority vote is in the cancellation of errors effect: when the errors of individual classifiers are independent or anticorrelated, and the error probability of individual classifiers is smaller than 0.5, then the errors average out, and the majority vote tends to outperform the individual classifiers (Masegosa et al., 2020). Furthermore, weighted majority vote remains an active topic in PAC-Bayesian research (Lorenzen et al., 2019; Wu et al., 2021; Masegosa et al., 2020; Viallard et al., 2021; Abbas & Andreopoulos, 2022; Wu & Seldin, 2022; Viallard et al., 2024; Hennequin et al., 2025).

Given a finite set of $n$ base classifiers $\mathcal{F} = \{\mathbf{f}_1, \dots, \mathbf{f}_n\}$, where $\mathbf{f}_i : \mathcal{X} \to \mathcal{Y}$, let $\mathcal{H}$ be the space of possible majority vote classifiers: $\mathcal{H} = \{\sum_{i=1}^n p_i \mathbf{f}_i \mid \mathbf{p} \in \mathcal{W}\}$, where $\mathcal{W} \subseteq \mathbb{R}^n$ is the majority vote weight space. We denote $\mathbf{h_w}(\mathbf{x}) = \sum_{i=1}^n w_i \mathbf{f}_i(\mathbf{x}) = \mathbf{w} \cdot \mathbf{f}(\mathbf{x}) \in \mathcal{Y}'$ the deterministic majority vote with weights $\mathbf{w}$. Since each majority vote is uniquely determined by its weight vector, we define $Q$ over the weight space $\mathcal{W}$, writing $\mathbf{w} \sim Q$ instead of $h_\mathbf{w} \sim Q$.

We now consider three families of posterior distributions, each parameterized by $\mathbf{p} \in \mathbb{R}^n$, which admit different sets of realizations $\mathcal{W}$: Categorical $\mathcal{C}(\mathbf{p})$, where $\mathcal{W} = \{\mathbf{w} \in \{0,1\}^n \mid \sum_{i=1}^n w_i = 1\}$; Dirichlet $D(\mathbf{p})$, where $\mathcal{W} = \{\mathbf{w} \in [0,1]^n \mid \sum_{i=1}^n w_i = 1\}$; and Unit-variance Gaussian $\mathcal{N}(\mathbf{p}, \mathbf{I}_{n\times n})$, where $\mathcal{W} = \mathbb{R}^n$.

For each distribution, we bound the risk of the single classifier corresponding to the average of the weight distribution $Q$. For example, for the Categorical distribution with parameters $\mathbf{p}$, we focus on $\mathbf{h_p}$. This choice is motivated by the fact that $R_{\mathcal{D}}(\mathbf{h_p})$ is most likely positively correlated with $R_{\mathcal{D}}(Q)$; if $R_{\mathcal{D}}(Q)$ is small, then $R_{\mathcal{D}}(\mathbf{h_p})$ will most likely be small as well.

### 4.1 Categorical assumption

The majority vote errs if at least half the total weight is assigned to base classifiers that make an error:

$$\ell(\mathbf{h_p}(\mathbf{x}), \mathbf{y}) = \mathbb{1}\left\{\sum_{i=1}^{n} p_i \mathbb{1}\left\{\mathbf{f}_i(\mathbf{x}) \neq \mathbf{y}\right\} \geq 0.5\right\}.$$

**Proposition 5** (Majority vote–Categorical derandomization)**.** *In the context of Proposition 1: let $Q = \mathcal{C}(\mathbf{p})$ be a Categorical distribution with parameters $\mathbf{p}$. For any data distribution $\mathcal{D}$ and $\mathbf{p} \in \{\mathbf{p}' \in [0,1]^n \mid \sum_{i=1}^{n} p_i' = 1\}$, we have*

$$\mathop{\mathbb{E}}_{\mathbf{w} \sim \mathcal{C}(\mathbf{p})} \ell(\mathbf{h_w}(\mathbf{x}), \mathbf{y}) = p_{\mathcal{F}},$$

$$\mathfrak{o}_{\mathcal{D}}^{\mathcal{C}(\mathbf{p})}(\mathbf{h_p}) = \mathop{\mathbb{E}}_{(\mathbf{x},\mathbf{y}) \sim \mathcal{D}} \left[p_{\mathcal{F}} \mid p_{\mathcal{F}} < 0.5\right],$$

$$\mathfrak{1}_{\mathcal{D}}^{\mathcal{C}(\mathbf{p})}(\mathbf{h_p}) = \mathop{\mathbb{E}}_{(\mathbf{x},\mathbf{y}) \sim \mathcal{D}} \left[p_{\mathcal{F}} \mid p_{\mathcal{F}} \geq 0.5\right],$$

*where $p_{\mathcal{F}} = \sum_{i=1}^{n} p_i \mathbb{1}\{\mathbf{f}_i(\mathbf{x}) \neq \mathbf{y}\}$.*

Note that $\mathfrak{1}_{\mathcal{D}}^{\mathcal{C}(\mathbf{p})}(\mathbf{h_p}) > \mathfrak{o}_{\mathcal{D}}^{\mathcal{C}(\mathbf{p})}(\mathbf{h_p})$ is ensured by their respective forms, which is a criterion to be met to use Corollary 4. Proposition 5 makes it explicit that $\mathfrak{1}_{\mathcal{D}}^{\mathcal{C}(\mathbf{p})}(\mathbf{h_p}) \geq 0.5$ and $\mathfrak{o}_{\mathcal{D}}^{\mathcal{C}(\mathbf{p})}(\mathbf{h_p}) \geq 0$, leading to the following worst-case scenario, when substituted in Proposition 1:

$$\forall \mathbf{h} \in \mathcal{H} \; : \; R_{\mathcal{D}}(\mathbf{h}) \leq \frac{R_{\mathcal{D}}(Q) - 0}{0.5 - 0} = 2R_{\mathcal{D}}(Q).$$

This recovers the classical "factor-2" bound from Langford & Shawe-Taylor (2002) as a worst case.

To obtain the best upper bound on $R_{\mathcal{D}}(\mathbf{h})$, we need the best deterministic lower bound on both $\mathfrak{o}_{\mathcal{D}}^{\mathcal{C}(\mathbf{p})}(\mathbf{h_p})$ and $\mathfrak{1}_{\mathcal{D}}^{\mathcal{C}(\mathbf{p})}(\mathbf{h_p})$. This in turn corresponds to finding the smallest possible value for $p_{\mathcal{F}}$, knowing $p_{\mathcal{F}}$ is respectively at least 0 and 0.5 (see Proposition 5). While the former, the smallest error it is possible to make, can usually be found in a reasonable computing time for majority vote classifiers, the latter is trickier to identify while remaining computationally tractable (there are $O(2^n)$ possible values for $p_{\mathcal{F}}$). To achieve this, we adopt the *partition problem* perspective.

**Definition** (The Partition Problem)**.** Given a set of non-negative numbers $\mathbf{a}$, the partition problem consists of finding

$$\operatorname*{argmin}_{\mathbf{a}_1, \mathbf{a}_2}\left\{\left|\sum_{a \in \mathbf{a}_1} a - \sum_{a \in \mathbf{a}_2} a\right| : \{\mathbf{a}_1, \mathbf{a}_2\} \text{ is a partition of } \mathbf{a}\right\}.$$

**Proposition 6** (Weights partitioning lower bound–Categorical)**.** *In the context of Proposition 5: let $\mathbf{p}_1$ and $\mathbf{p}_2$ be the result of the partition problem applied to $\mathbf{p}$. Then,*

$$\mathfrak{1}_{\mathcal{D}}^{\mathcal{C}(\mathbf{p})}(\mathbf{h_p}) \geq \max\left(\sum_{p \in \mathbf{p}_1} p, \sum_{p \in \mathbf{p}_2} p\right).$$

*Proof sketch.* The partition $\{\mathbf{p}_1, \mathbf{p}_2\}$ represents the worst-case scenario: voters' weights are split as evenly as possible between correct and incorrect predictions. Even in this worst case, when $\mathbf{h_p}$ makes an error (that is, when $p_{\mathcal{F}} \geq 0.5$), the weighted error fraction is at least $\max\left(\sum_{p \in \mathbf{p}_1} p, \sum_{p \in \mathbf{p}_2} p\right)$, providing the lower bound on $\mathfrak{1}_{\mathcal{D}}^{\mathcal{C}(\mathbf{p})}(\mathbf{h_p})$. □

The partition problem, while NP-complete, has been extensively studied: there exists a pseudo-polynomial time dynamic programming algorithm (Mertens, 2005), making the computation of our lower bound tractable.

### 4.2 Dirichlet assumption

We now consider the following loss function:

$$\ell(\mathbf{h_p}(\mathbf{x}), \mathbf{y}) = \mathbb{1}\left\{\sum_{i=1}^{n} p_i \mathbb{1}\left\{\mathbf{f}_i(\mathbf{x}) \neq \mathbf{y}\right\} \geq \frac{||\mathbf{p}||_1}{2}\right\}.$$

We consider the threshold $||\mathbf{p}||_1/2$ since Dirichlet parameters $\mathbf{p}$ can be unnormalized (not necessarily summing to 1); this naturally generalizes the Categorical case.

**Proposition 7** (Majority vote–Dirichlet derandomization). *In the context of Proposition 1: let $Q = D(\mathbf{p})$ be a Dirichlet distribution with parameters $\mathbf{p}$. For any data distribution $\mathcal{D}$ and $\mathbf{p} \in \mathbb{R}_{>0}^n$:*

$$\mathbb{E}_{\mathbf{w} \sim D(\mathbf{p})} \ell(\mathbf{h_w}(\mathbf{x}), \mathbf{y}) = I_{0.5}\big(||\mathbf{p}||_1 - p_{\mathcal{F}}, \, p_{\mathcal{F}}\big),$$

$$\mathfrak{o}_{\mathcal{D}}^{D(\mathbf{p})}(\mathbf{h_p}) = \mathbb{E}_{(\mathbf{x},\mathbf{y}) \sim \mathcal{D}} \left[ I_{0.5}\big(||\mathbf{p}||_1 - p_{\mathcal{F}}, \, p_{\mathcal{F}}\big)\, \Big|\, p_{\mathcal{F}} < \frac{||\mathbf{p}||_1}{2}\right],$$

$$\mathfrak{i}_{\mathcal{D}}^{D(\mathbf{p})}(\mathbf{h_p}) = \mathbb{E}_{(\mathbf{x},\mathbf{y}) \sim \mathcal{D}} \left[ I_{0.5}\big(||\mathbf{p}||_1 - p_{\mathcal{F}}, \, p_{\mathcal{F}}\big)\, \Big|\, p_{\mathcal{F}} \geq \frac{||\mathbf{p}||_1}{2}\right],$$

*where $p_{\mathcal{F}} = \sum_{i=1}^{n} p_i \mathbb{1}\{\mathbf{f}_i(\mathbf{x}) \neq \mathbf{y}\}$ and $I_x(\cdot, \cdot)$ is the regularized incomplete beta function evaluated at $x$.*

Notice, once again, that $\mathfrak{i}_{\mathcal{D}}^{D(\mathbf{p})}(\mathbf{h_p}) > \mathfrak{o}_{\mathcal{D}}^{D(\mathbf{p})}(\mathbf{h_p})$ is ensured since $I_{0.5}(\cdot, \cdot)$ is decreasing in its first argument, and increasing in its second one. We also have that $\mathfrak{i}_{\mathcal{D}}^{D(\mathbf{p})}(\mathbf{h_p}) \geq 0.5$ and $\mathfrak{o}_{\mathcal{D}}^{D(\mathbf{p})}(\mathbf{h_p}) \geq 0$, so that $R_{\mathcal{D}}(\mathbf{h}) \leq 2R_{\mathcal{D}}(Q)$ is the worst-case scenario. Finally, $\mathfrak{i}_{\mathcal{D}}^{D(\mathbf{p})}(\mathbf{h_p})$ can once again be lower-bounded in a way involving the partitioning problem, as per Proposition 8.

**Proposition 8** (Weights partitioning lower bound–Dirichlet). *In the context of Proposition 7: let $\mathbf{p}_1$ and $\mathbf{p}_2$ be the result of the partition problem applied to $\mathbf{p}$. Let $\widetilde{\mathbf{p}} = \max\left(\sum_{p \in \mathbf{p}_1} p, \sum_{p \in \mathbf{p}_2} p\right)$. Then,*

$$\mathfrak{i}_{\mathcal{D}}^{D(\mathbf{p})}(\mathbf{h_p}) \geq I_{0.5}\left(||\mathbf{p}||_1 - \widetilde{\mathbf{p}}, \, \widetilde{\mathbf{p}}\right).$$

As in the Categorical case, we lower-bound $\mathfrak{i}_{\mathcal{D}}^{D(\mathbf{p})}(\mathbf{h_p})$ by considering the most balanced partition of $\mathbf{p}$.

### 4.3 Gaussian assumption

We now consider the following loss function:

$$\ell(\mathbf{h_p}(\mathbf{x}), \mathbf{y}) = \mathbb{1}\left\{\mathbf{y} \neq \underset{\hat{\mathbf{y}} \in \mathcal{Y}}{\arg\max} \sum_{j=1}^{n} p_j \mathbb{1}\{\mathbf{f}_j(\mathbf{x}) = \hat{\mathbf{y}}\}\right\}.$$

We examine the binary classification and the multiclass classification setups independently, for they lead to different results.

**Binary classification.** Without loss of generality, let $\mathcal{Y} = \{-1, 1\}$.

**Proposition 9** (Majority vote–Binary Gaussian derandomization). *In the context of Proposition 1: let $Q = \mathcal{N}(\mathbf{p}, \mathbf{I})$ be a Gaussian distribution with mean $\mathbf{p}$ and identity covariance matrix. For any data distribution $\mathcal{D}$ and $\mathbf{p} \in \mathbb{R}^n$, we have*

$$\mathbb{E}_{\mathbf{w} \sim \mathcal{N}(\mathbf{p}, \mathbf{I})} \ell(\mathbf{h_w}(\mathbf{x}), y) = \Phi\left(y\frac{\mathbf{p} \cdot \mathbf{f}(\mathbf{x})}{||\mathbf{f}(\mathbf{x})||}\right),$$

$$\mathfrak{i}_{\mathcal{D}}^{\mathcal{N}(\mathbf{p}, \mathbf{I})}(\mathbf{h_p}) = 1 - \mathbb{E}_{(\mathbf{x},y) \sim \mathcal{D}}\left[\Phi\left(\frac{|\mathbf{p} \cdot \mathbf{f}(\mathbf{x})|}{||\mathbf{f}(\mathbf{x})||}\right)\, \Big|\, y(\mathbf{p} \cdot \mathbf{f}(\mathbf{x})) \leq 0\right],$$

$$\mathfrak{o}_{\mathcal{D}}^{\mathcal{N}(\mathbf{p}, \mathbf{I})}(\mathbf{h_p}) = \mathbb{E}_{(\mathbf{x},y) \sim \mathcal{D}}\left[\Phi\left(\frac{|\mathbf{p} \cdot \mathbf{f}(\mathbf{x})|}{||\mathbf{f}(\mathbf{x})||}\right)\, \Big|\, y(\mathbf{p} \cdot \mathbf{f}(\mathbf{x})) > 0\right],$$

with $\Phi(k) = \frac{1}{2}\left(1 - \text{erf}\left(\frac{k}{\sqrt{2}}\right)\right)$, $\text{erf}(k) = \frac{2}{\sqrt{\pi}}\int_0^k e^{-t^2}\,dt$.

Once again, both $R_{\mathcal{D}}(\mathbf{h}) \leq 2R_{\mathcal{D}}(Q)$ and $\mathbf{1}_{\mathcal{D}}^Q(\mathbf{h_p}) > \mathbf{o}_{\mathcal{D}}^Q(\mathbf{h_p})$ are ensured. Also, since every base classifier has a prediction in $\{-1, +1\}$, for every $\mathbf{x}$, we have $||\mathbf{f}(\mathbf{x})|| = \sqrt{n}$, we can lower-bound $\mathbf{1}_{\mathcal{D}}^Q(\mathbf{h_p})$ as in Sections 4.1 and 4.2.

**Proposition 10** (Weights partitioning lower bound–Binary Gaussian). *In the context of Proposition 9: let* $\mathbf{p}_1$ *and* $\mathbf{p}_2$ *be the result of the partition problem applied to* $\mathbf{p}$. *Let* $\overline{p} = \left|\sum_{p \in \mathbf{p}_1} p - \sum_{p \in \mathbf{p}_2} p\right|$. *Then,*

$$\mathbf{1}_{\mathcal{D}}^{\mathcal{N}(\mathbf{p},\mathrm{I})}(\mathbf{h_p}) \geq 1 - \Phi\left(\frac{\overline{p}}{\sqrt{n}}\right).$$

Unlike the Categorical and the Dirichlet cases, where the lower bound $\mathbf{o}_{\mathcal{D}}^Q(\mathbf{h_p}) \geq 0$ did not have a mathematical expression, the Gaussian structure allows a lower bound based on the partition problem.

**Proposition 11** (Weights maximizing lower bound–Binary Gaussian). *In the context of Proposition 9, we have*

$$\mathbf{o}_{\mathcal{D}}^{\mathcal{N}(\mathbf{p},\mathrm{I})}(\mathbf{h_p}) \geq \Phi\left(\frac{||\mathbf{p}||_1}{\sqrt{n}}\right).$$

**Multi-class classification.** In practice, Propositions 5, 7 and 9 are convenient provided that $\mathbb{E}_{\mathbf{h}' \sim Q}\ell(\mathbf{h}'(\mathbf{x}), \mathbf{y})$ can be easily computed for any $(\mathbf{x}, \mathbf{y}) \in \mathcal{X} \times \mathcal{Y}$. To our knowledge, an analytic expression exists in the PAC-Bayes literature for the case where $|\mathcal{Y}| = 2$ only (Langford & Shawe-Taylor, 2002), which we used in the derivation of Proposition 9. We generalize this result to the multi-class case by the following proposition. Note that $\mathbf{f}(\mathbf{x})$ is an $n \times k$ matrix (we recall that $k$ is the number of classes). Here, $\mathbf{f}_i(\mathbf{x})$ is the prediction of the $i^{\text{th}}$ base classifier, whereas $\mathbf{f}_{:,i}(\mathbf{x})$ is the $i^{\text{th}}$ column of the matrix.

**Proposition 12** (Majority vote–Multivariate Gaussian stochastic risk). *In the context of Proposition 1: let* $Q = \mathcal{N}(\mathbf{p}, \mathrm{I})$ *be a Gaussian distribution with mean* $\mathbf{p}$ *and identity covariance matrix. Let* $|\mathcal{Y}| = k$. *For any data distribution* $\mathcal{D}$ *and* $\mathbf{p} \in \mathbb{R}^n$:

$$\mathop{\mathbb{E}}_{\mathbf{w} \sim \mathcal{N}(\mathbf{p},\mathrm{I})} \ell(\mathbf{h_w}(\mathbf{x}), \mathbf{y}) = \sum_{i=1}^k \mathbb{1}\{y_i = 1\} F_{Z_i}(\mathbf{0}),$$

*where* $F$ *is the cumulative distribution function,* $Z_i \sim \mathcal{N}(\boldsymbol{\mu}_i, \Sigma_i)$ *is a* $(k-1)$-*variate Gaussian distribution with*

$$\mu_{i,j} = \begin{cases} \mathbf{p} \cdot (\mathbf{f}_{:,j}(\mathbf{x}) - \mathbf{f}_{:,i}(\mathbf{x})) & \text{if } j \in \{1, \dots, i-1\}, \\ \mathbf{p} \cdot (\mathbf{f}_{:,j+1}(\mathbf{x}) - \mathbf{f}_{:,i}(\mathbf{x})) & \text{if } j \in \{i, \dots, k-1\}, \end{cases}$$

$$\hat{\Sigma}_{i,j,k} = \begin{cases} (\mathbf{f}_{:,j}(\mathbf{x}) - \mathbf{f}_{:,i}(\mathbf{x})) \cdot (\mathbf{f}_{:,k}(\mathbf{x}) - \mathbf{f}_{:,i}(\mathbf{x})) & \text{if } j < i, k < i, \\ (\mathbf{f}_{:,j+1}(\mathbf{x}) - \mathbf{f}_{:,i}(\mathbf{x})) \cdot (\mathbf{f}_{:,k}(\mathbf{x}) - \mathbf{f}_{:,i}(\mathbf{x})) & \text{if } j \geq i, k < i, \\ (\mathbf{f}_{:,j}(\mathbf{x}) - \mathbf{f}_{:,i}(\mathbf{x})) \cdot (\mathbf{f}_{:,k+1}(\mathbf{x}) - \mathbf{f}_{:,i}(\mathbf{x})) & \text{if } j < i, k \geq i, \\ (\mathbf{f}_{:,j+1}(\mathbf{x}) - \mathbf{f}_{:,i}(\mathbf{x})) \cdot (\mathbf{f}_{:,k+1}(\mathbf{x}) - \mathbf{f}_{:,i}(\mathbf{x})) & \text{if } j \geq i, k \geq i. \end{cases}$$

Though we were not able to obtain an analytic expression for $\mathbf{o}_{\mathcal{D}}^{\mathcal{N}(\mathbf{p},\mathrm{I})}(\mathbf{h_p})$, $\mathbf{1}_{\mathcal{D}}^{\mathcal{N}(\mathbf{p},\mathrm{I})}(\mathbf{h_p})$ enabling lower-bounding analogous to Propositions 6, 8, 10 and 11, this result leads to a proper learning objective that several benchmarks can leverage, as shown in the following section.

## 5    Bound optimization

By deterministically lower-bounding $\mathbf{o}_{\mathcal{D}}^Q(\mathbf{h})$ and $\mathbf{1}_{\mathcal{D}}^Q(\mathbf{h})$ (Propositions 6, 8, 10 and 11), we are able to leverage Corollary 4. We call this method for bounding $R_{\mathcal{D}}(\mathbf{h})$ the *partition bound*.

While $R_{\mathcal{D}}(Q)$ can be estimated with any classical PAC-Bayes bound (e.g., Seeger, 2002), $\mathbf{1}_{\mathcal{D}}^Q(\mathbf{h})$ is bounded with Proposition 6 when $Q$ is Categorical, Proposition 8 when $Q$ is a Dirichlet distribution, and Proposition 10 when $Q$ is Gaussian and $k = 2$. The worst-case analysis for $\mathbf{o}_{\mathcal{D}}^Q(\mathbf{h})$ is done using Proposition 11 when $Q$ is Gaussian and $k = 2$, and manually otherwise.

**The partition problem computation.** The lower-bounding of $\mathbf{1}_{\mathcal{D}}^{Q}(\mathbf{h})$ relies on solving the partition problem, and while this problem is NP-complete, there exists a pseudo-polynomial time dynamic programming algorithm, and there are heuristics that solve the problem in many instances (Mertens, 2005). Such sets of values are found when the ratio "maximum number of bits to encode a single value" over "number of values to partition" is smaller than one (Mertens, 2001; Gent & Walsh, 2002). In the experimental section (Section 7), we consider values encoded with 32 bits and a number of base classifiers varying between 60 and 200 (ratio approximately in $[0.15, 0.5]$), leading to a time-efficient lower-bounding of $\mathbf{1}_{\mathcal{D}}^{Q}(\mathbf{h})$.

**Training and post-training heuristics.** During the training phase, we simply optimize the upper bound on the Gibbs risk $R_{\mathcal{D}}(Q)$ given by Theorem 13. Then, we apply several heuristics to tighten the partition bound. This separation is necessary because the deterministic bounding of $\mathbf{o}_{\mathcal{D}}^{Q}(\mathbf{h})$ and $\mathbf{1}_{\mathcal{D}}^{Q}(\mathbf{h})$ is non-differentiable.

Propositions 6, 8, 10 and 11 tells us that the bigger $\tilde{p} = \left| \sum_{p \in \mathbf{p}_1} p - \sum_{p \in \mathbf{p}_2} p \right|$, given that $\mathbf{p}_1$ and $\mathbf{p}_2$ are the result of the partition problem applied to $\mathbf{p}$, the tighter the bound. Thus, if no two partitions of $\mathbf{p}$ have similar total values, the better the partition bound. Leveraging this knowledge, after the optimization of $R_{\mathcal{D}}(Q)$, we apply three heuristics to improve the partition bound, each applied iteratively until no further improvement is achieved:

**Heuristic 1.** *We clip the smallest absolute values of $\mathbf{p}$ to 0, increasing $\tilde{p}$, until the upper bound from Corollary 4 is no longer lowered.*

**Heuristic 2.** *We apply a coordinate descent on the posterior values (increasing the largest components of $\mathbf{p}$ and decreasing the smallest ones), directly increasing $\tilde{p}$, until the upper bound from Corollary 4 does not lower anymore.*

**Heuristic 3.** *Since $\tilde{p}$ grows linearly with $||\mathbf{p}||_1$, we rescale the L1-norm $||\mathbf{p}||_1$ when it allows to reduce the partition bound value.*

## 6 Related works

### 6.1 Existing derandomized approaches via the Bayes classifier

We now present the main strategy from the PAC-Bayes literature for bounding the deterministic risk. We first need to define the so-called Bayes risk[2]; that is, the risk of the Bayes classifier:

$$\mathcal{B}_{\mathcal{D}}(Q) = \mathop{\mathbb{E}}_{(\mathbf{x},\mathbf{y}) \sim \mathcal{D}} \left[ \ell \left( \mathop{\mathbb{E}}_{\mathbf{h}' \sim Q} \mathbf{h}'(\mathbf{x}), \mathbf{y} \right) \right].$$

The approaches described below rely on bounding the Bayes risk by a function $f$ of the Gibbs risk:

$$\mathcal{B}_{\mathcal{D}}(Q) \leq f\left( R_{\mathcal{D}}(Q), \dots \right).$$

These approaches for bounding the risk of a single classifier rely on derandomization bounds for the Bayes classifier, which are consistent with the framework we developed. Our approach is more general and practical, in that it allows any hypothesis $\mathbf{h}$'s risk to be upper bounded, not only the Bayes classifier's, which usually implies restricting to models with very specific structure, such as in Letarte et al. (2019); Biggs & Guedj (2021).

While many papers discuss ways to bound the Bayes risk (Langford & Shawe-Taylor, 2002; Lacasse et al., 2010; Roy et al., 2011; Germain et al., 2015; Laviolette et al., 2017; Masegosa et al., 2020; Zantedeschi et al., 2021; Wu et al., 2021; Viallard et al., 2021), we focus on four of the most influential ones, using

$$\mathcal{I}_{\mathcal{D}}^{(k)} = \mathop{\mathbb{E}}_{(\mathbf{x},\mathbf{y}) \sim \mathcal{D}} \mathop{\mathbb{E}}_{\mathbf{h}_1, \dots, \mathbf{h}_k \sim Q} \mathbb{1} \left\{ \bigwedge_{j=1}^{k} \left[ \ell\left( \mathbf{h}_j(\mathbf{x}), \mathbf{y} \right) = 1 \right] \right\}, \ k \in \mathbb{N}_{>0}.$$

---

[2]The PAC-Bayes definition of the Bayes risk, which we adopt, is different from the risk of the Bayes-optimal predictor from the Bayesian literature.

**First-order bound.** The so-called *factor-2* bound (Langford & Shawe-Taylor, 2002) might be the most classical one. Using Markov's inequality, we obtain

$$\mathcal{B}_{\mathcal{D}}(Q) \leq 2\, \mathcal{I}_{\mathcal{D}}^{(1)} = 2R_{\mathcal{D}}(Q).$$

The bound considers the individual performance of the hypotheses independently, ignoring their correlation.

**Second-order bound.** To address this, Masegosa et al. (2020) focuses on improving the bounds by accounting for voter correlations, i.e., considering the agreement and/or disagreement of two random voters:

$$\mathcal{B}_{\mathcal{D}}(Q) \leq 4\, \mathcal{I}_{\mathcal{D}}^{(2)}.$$

**Binomial bound.** A generalization of the first-order approach was proposed in Shawe-Taylor & Hardoon (2009); Lacasse et al. (2010), where the Bayes risk is estimated by drawing multiple ($N$) hypotheses and computing the probability that at least half of them make an error:

$$\mathcal{B}_{\mathcal{D}}(Q) \leq 2\sum_{j=\frac{N}{2}}^{N} \binom{N}{k}\mathcal{I}_{\mathcal{D}}^{(j)}\,(1-\mathcal{I}_{\mathcal{D}})^{N-j}\,.$$

**C-bound.** Proposed by Lacasse et al. (2006), this bound is derived by considering explicitly the joint error and disagreement between two base predictors. Using Chebyshev-Cantelli's inequality:

$$\mathcal{B}_{\mathcal{D}}(Q) \leq \frac{\mathcal{I}_{\mathcal{D}}^{(2)} - \underset{\mathbf{h}'\sim Q}{\mathbb{E}}\big(R_{\mathcal{D}}(\mathbf{h}')\big)^2}{\mathcal{I}_{\mathcal{D}}^{(2)} - \mathcal{I}_{\mathcal{D}}^{(1)} + \frac{1}{4}}.$$

These methods rely on estimating any used $\mathcal{I}_{\mathcal{D}}^{(k)}$ with PAC-Bayes bounds. Optimizing the C-bound is a research topic of its own (Viallard et al., 2021).

## 6.2 Other derandomization approaches from the literature

A notable derivation of the PAC-Bayesian framework is found in disintegrated PAC-Bayes bounds (Blanchard & Fleuret, 2007; Rivasplata et al., 2020; Viallard et al., 2024; Clerico et al., 2025), which bound the risk of any classifier *randomly* drawn according to the posterior distribution $Q$. Such bounds apply to the generalization risk of a single classifier, yet this classifier is randomly drawn from the posterior distribution rather than arbitrarily chosen.

Other methods for derandomization involve particular loss functions (losses involving a *margin* parameter (Biggs & Guedj, 2022; Banerjee et al., 2020), or sub-gaussian losses (Banerjee et al., 2020)) or hypothesis architecture (Neyshabur et al., 2018; Nagarajan & Kolter, 2019).

## 6.3 A baseline based on the VC-dimension

Finally, we consider an alternative paradigm for bounding the deterministic risk of hypotheses that does not involve PAC-Bayes: the VC-dimension (VC-dim) (Vapnik, 2000), which is a measure of the expressive power of hypothesis sets. For any hypothesis set of finite VC-dim $v$, Vapnik (2000) showed that given $\delta \in [0,1]$, we have, with probability at least $1 - \delta$,

$$R_{\mathcal{D}}(\mathbf{h}) \leq \widehat{\mathcal{L}}_S(\mathbf{h}) + \sqrt{\frac{v\left(\ln\left(\frac{2m}{v}\right)+1\right) + \ln\left(\frac{4}{\delta}\right)}{m}}\,.$$

It is well-known (Vapnik, 2000) that for the class of weighted majority votes involving $n$ fixed base classifiers, i.e., the class of homogenous halfspaces in $n$ dimensions, the corresponding VC-dim is also $n$.

| $k$ | Distribution | First Order | Second Order | Binomial Bound | C-Bound | **Partion Bound** |
|---|---|---|---|---|---|---|
| $k = 2$ | Categorical | ✓ | ✓ | ✓ | ✓ | ★ |
| | Dirichlet | × | × | × | × | ★ |
| | Gaussian | ✓ | ✓ | ✓ | × | ★ |
| $k > 2$ | Categorical | ✓ | ✓ | ✓ | × | ★ |
| | Dirichlet | × | × | × | × | ★ |
| | Gaussian | ★ | ★ | ★ | × | × |

Table 1: Applicability of the true risk bounding method to derandomized majority vote, for binary ($k = 2$) and multi-class ($k > 2$) classification problems. **Green** (★): applicable, contribution; **lime** (✓): applicable, literature result; **red** (×): non-applicable.

### 6.4 Contributions contextualization

We show in Table 1 the applicability of the various ways to bound deterministic risks using stochastic risk found in the literature, and regarding our proposed approaches. We emphasize that no method from the literature permits the use of the Dirichlet distribution, and that we provide an objective for many baselines to use the Gaussian distribution for multi-class tasks.

## 7 Numerical experiments

We compare our proposed approach for bounding deterministic risk, the partition bound (Part), to all of the baselines discussed in Section 6: First-order, Second-order, Binomial, the C-Bound, and finally, the VC-dim bound. We experiment in two different settings[3].

In the first setting, we consider binary classification tasks and majority votes of data-independent hypotheses consisting of axis-aligned decision stumps, with thresholds evenly spread over the input space (10 per feature). The PAC-Bayes bound used for bounding $\mathcal{I}_{\mathcal{D}}^{(k)}$ is Seeger's bound (Seeger, 2002) with Maurer's trick (Maurer, 2004) (see Appendix A).

In the second setting, we consider multi-class classification tasks and majority votes of data-dependent hypotheses, consisting of a Random Forest (Breiman, 2001) of 200 trees as a set of voters $T$, sampling $\sqrt{d}$ random features to ensure voter diversity, optimizing Gini impurity score without bounding their maximal depth. To obtain a PAC-Bayes bound that allows learning the hypotheses on the training data, we rely on the following scheme: we split the training set $S$ into two sets $S_1$ and $S_2$; we learn half the trees $T_1$ of the random forest on $S_1$, the other half $T_2$ on $S_2$ so that $T_1 \cup T_2 = T$; we use a PAC-Bayes bound from Zantedeschi et al. (2021) to bound the expected risk of the resulting stochastic majority vote, requiring that we evaluate $T_1$ on $S_2$, $T_2$ on $S_1$. For the VC-dim-based approach, we simply reserved half the data for the learning of the forests and the other half for the majority vote weighting. See Section A for more details.

We experimented, for every baseline and our approaches, with every permissible distribution (see Table 1) and reported the results given by the best bound attained by a single distribution. We consider several classification datasets from various sources (see Section C for a complete datasets description). In the first setting, we considered the following tasks: ADULT, CODRNA, HABER, MUSH, PHIS, SVMG, TTT; in the second, we considered the following: FASHION, MNIST, PEND, PROTEIN, SENSOR.

We train the models by Stochastic Gradient Descent (SGD) using Adam (Kingma & Ba, 2015). Though a generalization of the C-Bound baseline to multi-class tasks has been proposed (Laviolette et al., 2014), no learning algorithm is provided: we only use this baseline on binary class tasks and use Algorithm 3 of Viallard et al. (2021). The objective function for the other PAC-Bayesian baselines corresponds to their bounds, whereas the VC-dim-based bound has its training cross-entropy loss as its learning objective. See Section C for more details on the experiments.

Table 2: Comparison of our proposed approach and a few baselines for different types of predictors. **Bolded** and underlined values are respectively within one standard deviation of the best bound and test error values. Average and standard deviations computed over 5 random seeds.

| Task | First Order | | Second Order | | Binomial Bound | | C-Bound | | VC-dimension | | **Partition Bound** | |
|---|---|---|---|---|---|---|---|---|---|---|---|---|
| | Bound | Test error | Bound | Test error | Bound | Test error | Bound | Test error | Bound | Test error | Bound | Test error |
| ADULT | $46.9 \pm 0.0$ | $22.2 \pm 0.1$ | $52.9 \pm 0.1$ | $\underline{15.8} \pm 0.2$ | $51.4 \pm 0.0$ | $24.1 \pm 0.0$ | $74.5 \pm 0.4$ | $21.0 \pm 1.6$ | $78.9 \pm 0.0$ | $17.2 \pm 0.3$ | $\mathbf{23.2} \pm 2.4$ | $21.7 \pm 2.5$ |
| CODRNA | $40.2 \pm 0.1$ | $12.0 \pm 0.1$ | $47.4 \pm 0.1$ | $\underline{11.7} \pm 0.1$ | $42.8 \pm 5.2$ | $15.2 \pm 4.5$ | $63.7 \pm 1.0$ | $23.1 \pm 1.3$ | $38.8 \pm 1.6$ | $23.1 \pm 1.5$ | $\mathbf{24.7} \pm 0.3$ | $23.6 \pm 0.3$ |
| HABER | $75.2 \pm 0.9$ | $\underline{26.5} \pm 1.8$ | $110.5 \pm 1.1$ | $26.1 \pm 2.1$ | $84.7 \pm 0.8$ | $\underline{26.1} \pm 0.7$ | $100.0 \pm 0.0$ | $\underline{27.1} \pm 1.8$ | $112.0 \pm 1.0$ | $\underline{28.4} \pm 1.8$ | $\mathbf{37.8} \pm 0.5$ | $26.5 \pm 1.8$ |
| MUSH | $12.2 \pm 0.3$ | $4.8 \pm 0.5$ | $20.6 \pm 1.2$ | $\underline{0.5} \pm 0.3$ | $12.6 \pm 0.4$ | $1.1 \pm 0.1$ | $22.0 \pm 0.5$ | $4.8 \pm 0.5$ | $59.1 \pm 0.1$ | $4.8 \pm 0.5$ | $\mathbf{6.1} \pm 0.1$ | $4.8 \pm 0.5$ |
| PHIS | $25.8 \pm 0.3$ | $11.1 \pm 0.6$ | $33.6 \pm 0.2$ | $\underline{6.4} \pm 0.5$ | $27.5 \pm 0.2$ | $\underline{6.8} \pm 0.5$ | $37.1 \pm 0.5$ | $10.4 \pm 1.3$ | $85.2 \pm 0.2$ | $11.1 \pm 0.6$ | $\mathbf{24.9} \pm 1.0$ | $11.1 \pm 0.6$ |
| SVMG | $17.4 \pm 0.6$ | $\underline{7.0} \pm 1.0$ | $28.2 \pm 0.7$ | $\underline{7.0} \pm 1.0$ | $22.0 \pm 0.6$ | $\underline{7.0} \pm 1.0$ | $29.2 \pm 0.7$ | $\underline{7.0} \pm 1.0$ | $37.6 \pm 0.3$ | $\underline{7.0} \pm 1.0$ | $\mathbf{8.7} \pm 0.3$ | $\underline{7.0} \pm 1.0$ |
| TTT | $75.9 \pm 1.9$ | $\underline{30.4} \pm 3.8$ | $100.0 \pm 0.7$ | $\underline{30.2} \pm 2.7$ | $87.2 \pm 1.2$ | $29.5 \pm 4.3$ | $94.8 \pm 0.7$ | $\underline{30.4} \pm 3.8$ | $117.0 \pm 1.8$ | $\underline{32.2} \pm 2.9$ | $\mathbf{48.4} \pm 12.6$ | $\underline{30.4} \pm 3.8$ |
| FASHION | $40.7 \pm 0.2$ | $17.1 \pm 0.2$ | $58.6 \pm 0.2$ | $\underline{13.0} \pm 0.3$ | $49.9 \pm 0.3$ | $\underline{13.0} \pm 0.3$ | N/A | N/A | $40.3 \pm 0.3$ | $18.3 \pm 0.3$ | $\mathbf{26.6} \pm 0.1$ | $25.4 \pm 0.2$ |
| MNIST | $26.9 \pm 0.1$ | $8.0 \pm 0.4$ | $44.7 \pm 0.1$ | $\underline{4.4} \pm 0.1$ | $35.8 \pm 0.2$ | $\underline{4.3} \pm 0.2$ | N/A | N/A | $37.4 \pm 2.6$ | $15.8 \pm 2.7$ | $\mathbf{23.6} \pm 0.2$ | $22.6 \pm 0.3$ |
| PEND | $12.0 \pm 0.2$ | $2.5 \pm 0.3$ | $17.1 \pm 0.1$ | $\underline{1.5} \pm 0.2$ | $15.5 \pm 0.2$ | $1.5 \pm 0.2$ | N/A | N/A | $52.7 \pm 0.6$ | $5.8 \pm 0.7$ | $\mathbf{8.9} \pm 0.2$ | $\underline{1.4} \pm 0.2$ |
| PROTEIN | $84.9 \pm 0.6$ | $\underline{34.9} \pm 0.3$ | $138.3 \pm 0.3$ | $36.6 \pm 0.5$ | $120.1 \pm 1.3$ | $55.0 \pm 0.5$ | N/A | N/A | $73.4 \pm 0.3$ | $39.7 \pm 0.5$ | $\mathbf{55.8} \pm 0.4$ | $53.8 \pm 0.6$ |
| SENSOR | $16.8 \pm 0.4$ | $4.1 \pm 0.3$ | $21.2 \pm 1.0$ | $3.9 \pm 0.4$ | $15.3 \pm 1.0$ | $3.0 \pm 0.2$ | N/A | N/A | $32.9 \pm 2.0$ | $9.6 \pm 2.0$ | $\mathbf{1.4} \pm 0.1$ | $\underline{0.4} \pm 0.2$ |

Table 3: Proposed approach's bound and test error for every distribution. **Bolded** and underlined values are respectively within one standard deviation of the best bound and test error values. Average and standard deviations computed over 5 random seeds.

| Task | Categorical Distribution | | Dirichlet Distribution | | Gaussian Distribution | |
|---|---|---|---|---|---|---|
| | Partition Bound | Test error | Partition Bound | Test error | Partition Bound | Test error |
| ADULT | $46.4 \pm 0.4$ | $22.2 \pm 0.1$ | $\mathbf{23.2} \pm 2.4$ | $\underline{21.7} \pm 2.5$ | $51.0 \pm 0.0$ | $24.1 \pm 0.0$ |
| CODRNA | $\mathbf{24.7} \pm 0.3$ | $23.6 \pm 0.3$ | $29.4 \pm 4.8$ | $28.4 \pm 4.6$ | $39.1 \pm 1.4$ | $\underline{17.1} \pm 7.2$ |
| HABER | $\mathbf{37.8} \pm 0.5$ | $\underline{26.5} \pm 1.8$ | $73.6 \pm 7.9$ | $34.2 \pm 7.2$ | $71.7 \pm 4.2$ | $\underline{25.8} \pm 3.0$ |
| MUSH | $\mathbf{6.1} \pm 0.1$ | $4.8 \pm 0.5$ | $8.0 \pm 1.0$ | $\underline{4.1} \pm 1.8$ | $19.0 \pm 0.5$ | $\underline{2.6} \pm 0.6$ |
| PHIS | $\mathbf{24.9} \pm 1.0$ | $11.1 \pm 0.6$ | $26.8 \pm 0.2$ | $\underline{6.7} \pm 0.4$ | $27.5 \pm 0.2$ | $\underline{6.6} \pm 0.5$ |
| SVMG | $\mathbf{8.7} \pm 0.3$ | $7.0 \pm 1.0$ | $\mathbf{9.5} \pm 1.7$ | $\underline{5.3} \pm 1.4$ | $14.0 \pm 0.2$ | $7.0 \pm 1.0$ |
| TTT | $\mathbf{48.4} \pm 12.6$ | $30.4 \pm 3.8$ | $84.3 \pm 1.3$ | $\underline{29.9} \pm 4.1$ | $82.7 \pm 9.5$ | $36.5 \pm 6.7$ |
| FASHION | $\mathbf{26.6} \pm 0.1$ | $25.4 \pm 0.2$ | $44.8 \pm 0.3$ | $\underline{13.0} \pm 0.3$ | N/A | N/A |
| MNIST | $\mathbf{23.6} \pm 0.2$ | $22.6 \pm 0.3$ | $31.3 \pm 0.2$ | $\underline{4.3} \pm 0.2$ | N/A | N/A |
| PEND | $10.6 \pm 0.2$ | $8.3 \pm 0.7$ | $\mathbf{8.9} \pm 0.2$ | $\underline{1.4} \pm 0.2$ | N/A | N/A |
| PROTEIN | $\mathbf{55.8} \pm 0.4$ | $\underline{53.8} \pm 0.6$ | $55.8 \pm 3.3$ | $54.9 \pm 0.5$ | N/A | N/A |
| SENSOR | $3.4 \pm 0.3$ | $2.9 \pm 0.3$ | $\mathbf{1.4} \pm 0.1$ | $\underline{0.4} \pm 0.2$ | N/A | N/A |

### 7.1 Discussion

Table 2 shows that, on a total of 12 datasets in the two different learning paradigms, the proposed approach's generalization bound is always at least as good as the baselines'. The VC-dim-based approach provides loose generalization bounds because the bound is penalized proportionally to the base classifier set size, even when some of these classifiers end up having a small role in the majority vote; the PAC-Bayes-based approaches are more robust to this issue by providing those with a weight similar to that of the prior distribution. In many cases, the proposed bound corresponds to about half the best runner-up bound (ADULT, CODRNA, SVMG, etc.) On five tasks out of twelve, the proposed approach even has a test error at most one standard deviation away from the best test value. Combined, these results show the dominance of our approach in many scenarios. We display in Table 3 the partition bound and test error obtained for each distribution. Even though the Categorical distribution definitely seems to have an advantage over both the Dirichlet and the Gaussian distribution when it comes to the generalization bound, the latter two distributions show impressive test error results, justifying the development and use of all three of them.

## 8 Conclusion

In this work, we developed a general framework for deriving, from PAC-Bayes, generalization guarantees holding for a single hypothesis: *Derandomization* bounds. We develop such an oracle bound before specializing it to majority votes with weighting involving either the Categorical, the Dirichlet, or the Gaussian

---

[3]The code to reproduce these experiments will be made public upon acceptance.

distribution. We test (and compare to several benchmarks) our approach on many binary and multi-class classification tasks, providing empirical evidence of the tightness of our approach.

Our derandomization framework is consistent with existing PAC-Bayesian bounding methods while being more general: it allows bounding the risk of any single classifier in $\mathcal{H}$, not only that of the Bayes classifier.

**Future works.** Lower bounds on the risk of a single hypothesis cannot be obtained using the methods surveyed in Section 6, yet they can be obtained through our derandomization framework, using, for example, Corollaries 3 and 4. Though lower bounds on risk metrics are less popular than upper ones, they appear necessary in bounding metrics involving the risk of a single hypothesis on various tasks, such as in fairness computation (Oneto et al., 2020). Therefore, a unique and practical application of our derandomization method could be found in bounding the value of such a metric for single classifiers.

There are several other directions for future work: extending beyond majority votes to neural networks, deriving multi-class Gaussian bounds, and developing end-to-end differentiable approximations to the partition problem. Also, a natural extension of this work lies in using any (possibly unbounded) loss $\ell : \mathcal{Y}' \times \mathcal{Y} \to \mathbb{R}_{\geq 0}$. We believe this could be achieved by defining $\mathfrak{o}_{\mathcal{D}}^{Q}(\mathbf{h})$ and $\mathfrak{1}_{\mathcal{D}}^{Q}(\mathbf{h})$ as such:

$$\mathfrak{o}_{\mathcal{D}}^{Q}(\mathbf{h}) = \mathop{\mathbb{E}}_{(\mathbf{x},\mathbf{y})\sim\mathcal{D}} \left[ \mathop{\mathbb{E}}_{\mathbf{h}'\sim Q} \ell(\mathbf{h}'(\mathbf{x}),\mathbf{y}) \;\middle|\; \ell(\mathbf{h}(\mathbf{x}),\mathbf{y}) \leq \gamma \right],$$

$$\mathfrak{1}_{\mathcal{D}}^{Q}(\mathbf{h}) = \mathop{\mathbb{E}}_{(\mathbf{x},\mathbf{y})\sim\mathcal{D}} \left[ \mathop{\mathbb{E}}_{\mathbf{h}'\sim Q} \ell(\mathbf{h}'(\mathbf{x}),\mathbf{y}) \;\middle|\; \ell(\mathbf{h}(\mathbf{x}),\mathbf{y}) > \gamma \right],$$

for some $\gamma \geq 0$. Adapting the loss function to reflect this $\gamma$ parameter and interpreting it as a *margin*-related parameter could enable utilizing margin-based generalization bounds (Gao & Zhou, 2013; Zantedeschi et al., 2021; Biggs et al., 2022; Biggs & Guedj, 2022) as building blocks in our derandomization framework, further tying up our approach with existing PAC-Bayes bounds from the literature.

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

## A  PAC-Bayes bounds from the literature

**Data-independent case.**

**Theorem 13.** (Seeger, 2002; Maurer, 2004) *Let* $\mathrm{kl}(q,p) = q\ln(\frac{q}{p}) + (1-q)\ln(\frac{1-q}{1-p})$. *For any distribution* $\mathcal{D}$, *hypothesis set* $\mathcal{H}$, *prior distribution* $P$ *over* $\mathcal{H}$, $\delta \in (0,1]$, *we have with probability at least* $1-\delta$ *over the random choice* $S \sim \mathcal{D}^m$ *that for every* $Q$ *over* $\mathcal{H}$:

$$\mathrm{kl}\left( R_S(Q) \middle\| R_\mathcal{D}(Q) \right) \leq \frac{1}{m}\left[ \mathrm{KL}(Q\|P) + \ln\left( \frac{2\sqrt{m}}{\delta} \right) \right].$$

**Data-dependent case.**     Here, we present a cross-bounding certificate (Zantedeschi et al., 2021) that allows us to learn and evaluate the set of base classifiers without held-out data. More precisely, we split the training data $S$ into two subsets ($S_{\leq j} = \{(\mathbf{x}_i, \mathbf{y}_i) \in S\}_{i=1}^j$ and $S_{>j} = \{(\mathbf{x}_i, \mathbf{y}_i) \in S\}_{i=j+1}^m$ for some $j$) and we learn a set of base classifiers on each data split independently (determining the hypothesis spaces $\mathcal{H}_{\leq j}$ and $\mathcal{H}_{>j}$). We refer to the prior distribution over $\mathcal{H}_{\leq j}$ as $P_{\leq j}$ and to the prior distribution over $\mathcal{H}_{>j}$ as $P_{>j}$. In the same way, we can then define a posterior distribution per hypothesis space: $Q_{\leq j}$ and $Q_{>j}$. The following theorem shows that we can bound the expected risk of any convex combination of the two posteriors, as long as their empirical risks are evaluated on the data split that was not used for learning their respective priors.

**Theorem 14.** (Zantedeschi et al., 2021) *Let* $\mathrm{kl}(q,p) = q\ln(\frac{q}{p}) + (1-q)\ln(\frac{1-q}{1-p})$. *For any distribution* $\mathcal{D}$, *hypothesis sets* $\mathcal{H}_{\leq j}$ *and* $\mathcal{H}_{>j}$, *prior distributions* $P_{\leq j}$ *over* $\mathcal{H}_{\leq j}$ *and* $P_{>j}$ *over* $\mathcal{H}_{>j}$, $\delta \in (0,1]$ *and* $\alpha \in [0,1]$, *we have with probability at least* $1-\delta$ *over* $S \sim \mathcal{D}^m$ *that for every* $Q_{\leq j}$ *over* $\mathcal{H}_{\leq j}$ *and* $Q_{>j}$ *over* $\mathcal{H}_{>j}$:

$$\mathrm{kl}\bigg( \alpha R_{S_{>j}}(Q_{\leq j}) + (1-\alpha)R_{S_{\leq j}}(Q_{>j}) \,\bigg\|\, \alpha R_\mathcal{D}(Q_{\leq j}) + (1-\alpha)R_\mathcal{D}(Q_{>j}) \bigg)$$

$$\leq \frac{\alpha \mathrm{KL}(Q_{>j}\|P_{>j})}{j} + \frac{(1-\alpha)\mathrm{KL}(Q_{\leq j}\|P_{\leq j})}{m-j} + \frac{1}{m}\ln\left( \frac{4\sqrt{j(m-j)}}{\delta} \right).$$

## B  Mathematical proofs

**Proposition 1** (Stochastic-deterministic relation)**.** *For any data distribution $\mathcal{D}$, distribution $Q$ over $\mathcal{H}$ and classifier $\mathbf{h} \in \mathcal{H}$ such that $\mathbf{1}_{\mathcal{D}}^{Q}(\mathbf{h}) \neq \mathbf{o}_{\mathcal{D}}^{Q}(\mathbf{h})$, we have*

$$R_{\mathcal{D}}(\mathbf{h}) = \frac{R_{\mathcal{D}}(Q) - \mathbf{o}_{\mathcal{D}}^{Q}(\mathbf{h})}{\mathbf{1}_{\mathcal{D}}^{Q}(\mathbf{h}) - \mathbf{o}_{\mathcal{D}}^{Q}(\mathbf{h})}.$$

*Proof.*

$$\mathop{\mathbb{E}}_{\mathbf{h}' \sim Q} R_{\mathcal{D}}(\mathbf{h}') = \mathop{\mathbb{E}}_{(\mathbf{x}, \mathbf{y}) \sim \mathcal{D}} \mathop{\mathbb{E}}_{\mathbf{h}' \sim Q} \ell(\mathbf{h}'(\mathbf{x}), \mathbf{y})$$

$$= \mathop{\mathbb{P}}_{(\mathbf{x}, \mathbf{y}) \sim \mathcal{D}} \Big( \ell(\mathbf{h}(\mathbf{x}), \mathbf{y}) = 1 \Big) \mathop{\mathbb{E}}_{(\mathbf{x}, \mathbf{y}) \sim \mathcal{D}} \left[ \mathop{\mathbb{E}}_{\mathbf{h}' \sim Q} \ell(\mathbf{h}'(\mathbf{x}), \mathbf{y}) \,\middle|\, \ell(\mathbf{h}(\mathbf{x}), \mathbf{y}) = 1 \right] +$$

$$\mathop{\mathbb{P}}_{(\mathbf{x}, \mathbf{y}) \sim \mathcal{D}} \Big( \ell(\mathbf{h}(\mathbf{x}), \mathbf{y}) = 0 \Big) \mathop{\mathbb{E}}_{(\mathbf{x}, \mathbf{y}) \sim \mathcal{D}} \left[ \mathop{\mathbb{E}}_{\mathbf{h}' \sim Q} \ell(\mathbf{h}'(\mathbf{x}), \mathbf{y}) \,\middle|\, \ell(\mathbf{h}(\mathbf{x}), \mathbf{y}) = 0 \right]$$

$$= \mathbf{1}_{\mathcal{D}}^{Q}(\mathbf{h}) \mathop{\mathbb{P}}_{(\mathbf{x}, \mathbf{y}) \sim \mathcal{D}} \Big( \ell(\mathbf{h}(\mathbf{x}), \mathbf{y}) = 1 \Big) + \mathbf{o}_{\mathcal{D}}^{Q}(\mathbf{h}) \mathop{\mathbb{P}}_{(\mathbf{x}, \mathbf{y}) \sim \mathcal{D}} \Big( \ell(\mathbf{h}(\mathbf{x}), \mathbf{y}) = 0 \Big)$$

$$= \mathbf{1}_{\mathcal{D}}^{Q}(\mathbf{h}) \mathop{\mathbb{P}}_{(\mathbf{x}, \mathbf{y}) \sim \mathcal{D}} \Big( \ell(\mathbf{h}(\mathbf{x}), \mathbf{y}) = 1 \Big) + \mathbf{o}_{\mathcal{D}}^{Q}(\mathbf{h}) \Big( 1 - \mathop{\mathbb{P}}_{(\mathbf{x}, \mathbf{y}) \sim \mathcal{D}} \Big( \ell(\mathbf{h}(\mathbf{x}), \mathbf{y}) = 1 \Big) \Big)$$

$$= (\mathbf{1}_{\mathcal{D}}^{Q}(\mathbf{h}) - \mathbf{o}_{\mathcal{D}}^{Q}(\mathbf{h})) \mathop{\mathbb{P}}_{(\mathbf{x}, \mathbf{y}) \sim \mathcal{D}} \Big( \ell(\mathbf{h}(\mathbf{x}), \mathbf{y}) = 1 \Big) + \mathbf{o}_{\mathcal{D}}^{Q}(\mathbf{h})$$

$$= (\mathbf{1}_{\mathcal{D}}^{Q}(\mathbf{h}) - \mathbf{o}_{\mathcal{D}}^{Q}(\mathbf{h})) \mathop{\mathbb{E}}_{(\mathbf{x}, \mathbf{y}) \sim \mathcal{D}} \ell(\mathbf{h}(\mathbf{x}), \mathbf{y}) + \mathbf{o}_{\mathcal{D}}^{Q}(\mathbf{h})$$

$$= (\mathbf{1}_{\mathcal{D}}^{Q}(\mathbf{h}) - \mathbf{o}_{\mathcal{D}}^{Q}(\mathbf{h})) R_{\mathcal{D}}(\mathbf{h}) + \mathbf{o}_{\mathcal{D}}^{Q}(\mathbf{h}) \,.$$

The main result is obtained by a simple rearranging of the terms, given $\mathbf{1}_{\mathcal{D}}^{Q}(\mathbf{h}) - \mathbf{o}_{\mathcal{D}}^{Q}(\mathbf{h}) \neq 0$. $\qquad\square$

**Lemma 15.** (Maurer, 2004) *For a distribution $\mathcal{D}$ over $\mathcal{X} \times \mathcal{Y}$, a loss function $\ell : \mathcal{Y}' \times \mathcal{Y} \to [0, 1]$, a hypothesis $\mathbf{h} : \mathcal{X} \to \mathcal{Y}'$, where $\mathbf{h} \in \mathcal{H}$, and a convex function $\kappa : [0, 1] \to \mathbb{R}$:*

$$\mathop{\mathbb{E}}_{S \sim \mathcal{D}^m} \kappa\left(R_S(\mathbf{h})\right) \leq \sum_{i=0}^{m} \binom{m}{i} \left(R_{\mathcal{D}}(\mathbf{h})\right)^i \left(1 - R_{\mathcal{D}}(\mathbf{h})\right)^{m-i} \kappa\left(\frac{i}{m}\right).$$

**Theorem 2** (Conditional PAC-Bayes)**.** *Let $\mathrm{kl}(q, p) = q \ln\left(\frac{q}{p}\right) + (1-q) \ln\left(\frac{1-q}{1-p}\right)$. For any data distribution $\mathcal{D}$, hypothesis $\mathbf{h} \in \mathcal{H}$, prior distribution $\mathcal{P}$ over $\mathcal{H}$, and $\delta \in (0, 1]$, both the following statements hold with probability at least $1 - \delta$ over the draw $S \sim \mathcal{D}^m$, for all $Q$ over $\mathcal{H}$:*

$$\mathrm{kl}\left(\mathbf{o}_{S}^{Q}(\mathbf{h}) \,\middle\|\, \mathbf{o}_{\mathcal{D}}^{Q}(\mathbf{h})\right) \leq \frac{1}{m_{(\mathbf{h},0)}} \left[ \mathrm{KL}(Q\|P) + \ln\left(\frac{2\sqrt{m}}{\delta}\right) \right],$$

$$\mathrm{kl}\left(\mathbf{1}_{S}^{Q}(\mathbf{h}) \,\middle\|\, \mathbf{1}_{\mathcal{D}}^{Q}(\mathbf{h})\right) \leq \frac{1}{m_{(\mathbf{h},1)}} \left[ \mathrm{KL}(Q\|P) + \ln\left(\frac{2\sqrt{m}}{\delta}\right) \right],$$

*where $\mathbf{o}_{S}^{Q}(\mathbf{h}) = R_{S_{(\mathbf{h},0)}}(Q)$, $\mathbf{1}_{S}^{Q}(\mathbf{h}) = R_{S_{(\mathbf{h},1)}}(Q)$, $S_{(\mathbf{h},\cdot)} = \{(\mathbf{x}, \mathbf{y}) \in S : \ell(\mathbf{h}(\mathbf{x}), \mathbf{y}) = \cdot\}$, and $|S_{(\mathbf{h},\cdot)}| = m_{(\mathbf{h},\cdot)}$.*

*Proof.* Let us prove the first statement of the main result; the second one can be proved using similar manipulations. Given a posterior $Q$, let $\mathcal{D}_{(\mathbf{h},0)}$ be a distribution with probability density function

$$f_{\mathcal{D}_{(\mathbf{h},0)}}(\mathbf{x}, \mathbf{y}) \;=\; \frac{f_{\mathcal{D}}(\mathbf{x}, \mathbf{y}) \cdot \mathbb{1}\{\ell(\mathbf{h}(\mathbf{x}), \mathbf{y}) = 0\}}{\mathbb{P}_{\mathcal{D}}\big(\ell(\mathbf{h}(\mathbf{x}), \mathbf{y}) = 0\big)} \,.$$

Note that for any dataset $S \sim \mathcal{D}^m$ and hypothesis $\mathbf{h}$, any example from $S_{(\mathbf{h},0)}$ can be seen as a realization from $\mathcal{D}_{(\mathbf{h},0)}$.

We want to upper bound $\mathrm{kl}\left(\mathrm{o}_S^Q(\mathbf{h})\middle\|\mathrm{o}_\mathcal{D}^Q(\mathbf{h})\right)$ for every posterior $Q$. Given a dataset $S$, we have

$$\forall Q \text{ over } \mathcal{H} \; : \qquad |S_{(\mathbf{h},0)}| \cdot \mathrm{kl}\left(\mathrm{o}_S^Q(\mathbf{h})\middle\|\mathrm{o}_\mathcal{D}^Q(\mathbf{h})\right)$$

$$= |S_{(\mathbf{h},0)}| \cdot \mathrm{kl}\left(\mathop{\mathbb{E}}_{\mathbf{h}'\sim Q} R_{S_{(\mathbf{h},0)}}(\mathbf{h}')\middle\|\mathop{\mathbb{E}}_{\mathbf{h}'\sim Q} R_{\mathcal{D}_{(\mathbf{h},0)}}(\mathbf{h}')\right)$$

$$\leq |S_{(\mathbf{h},0)}| \mathop{\mathbb{E}}_{\mathbf{h}'\sim Q} \mathrm{kl}\left(R_{S_{(\mathbf{h},0)}}(\mathbf{h}')\middle\|R_{\mathcal{D}_{(\mathbf{h},0)}}(\mathbf{h}')\right) \quad \langle\text{Jenson's inequality (kl is convex)}\rangle$$

$$\leq \mathrm{KL}(Q\|P) + \ln\left(\mathop{\mathbb{E}}_{\mathbf{h}'\sim P} e^{|S_{(\mathbf{h},0)}|\cdot\mathrm{kl}\left(R_{S_{(\mathbf{h},0)}}(\mathbf{h}')\middle\|R_{\mathcal{D}_{(\mathbf{h},0)}}(\mathbf{h}')\right)}\right). \quad \langle\text{Change of measure}\rangle$$

Let's now consider the random variable, with respect to $S$:

$$X_{\mathbf{h},\mathcal{H},S,P} = \mathbb{E}_{\mathbf{h}'\sim P}\exp\left(|S_{\mathbf{h}}| \cdot \mathrm{kl}\left(R_{S_{(\mathbf{h},0)}}(\mathbf{h}')\middle\|R_{\mathcal{D}_{(\mathbf{h},0)}}(\mathbf{h}')\right)\right).$$

With Markov's inequality, we have

$$\mathop{\mathbb{P}}_{S\sim\mathcal{D}^m}\left(X_{\mathbf{h},\mathcal{H},S,P} \leq \frac{1}{\delta}\mathop{\mathbb{E}}_{S'\sim\mathcal{D}^m} X_{\mathbf{h},\mathcal{H},S',P}\right) \geq 1-\delta.$$

Thus, with probability at least $1-\delta$ over $S\sim\mathcal{D}^m$:

$$\forall Q \text{ over } \mathcal{H} \; : \; |S_{(\mathbf{h},0)}| \cdot \mathrm{kl}\left(\mathrm{o}_S^Q(\mathbf{h})\middle\|\mathrm{o}_\mathcal{D}^Q(\mathbf{h})\right) \leq \mathrm{KL}(Q\|P) + \ln\left(\frac{1}{\delta}\mathop{\mathbb{E}}_{S'\sim\mathcal{D}^m} X_{\mathbf{h},\mathcal{H},S',P}\right).$$

We bound $\mathop{\mathbb{E}}_{S'\sim\mathcal{D}^m} X_{\mathbf{h},\mathcal{H},S',P}$ as such:

$$\mathop{\mathbb{E}}_{S'\sim\mathcal{D}^m} X_{\mathbf{h},\mathcal{H},S',P}$$

$$= \mathop{\mathbb{E}}_{S'\sim\mathcal{D}^m} \mathop{\mathbb{E}}_{\mathbf{h}'\sim P} e^{|S'_{(\mathbf{h},0)}|\cdot\mathrm{kl}\left(R_{S'_{(\mathbf{h},0)}}(\mathbf{h}')\middle\|R_{\mathcal{D}_{(\mathbf{h},0)}}(\mathbf{h}')\right)}$$

$$= \mathop{\mathbb{E}}_{\mathbf{h}'\sim P} \mathop{\mathbb{E}}_{S'\sim\mathcal{D}^m} e^{|S'_{(\mathbf{h},0)}|\cdot\mathrm{kl}\left(R_{S'_{(\mathbf{h},0)}}(\mathbf{h}')\middle\|R_{\mathcal{D}_{(\mathbf{h},0)}}(\mathbf{h}')\right)}$$

$$= \mathop{\mathbb{E}}_{\mathbf{h}'\sim P} \sum_{i=0}^{m} \mathop{\mathbb{P}}_{S'\sim\mathcal{D}^m}\left(|S'_{(\mathbf{h},0)}|=i\right) \mathop{\mathbb{E}}_{S'\sim\mathcal{D}^m}\left[e^{|S'_{(\mathbf{h},0)}|\cdot\mathrm{kl}\left(R_{S'_{(\mathbf{h},0)}}(\mathbf{h}')\middle\|R_{\mathcal{D}_{(\mathbf{h},0)}}(\mathbf{h}')\right)} \;\middle|\; |S'_{(\mathbf{h},0)}|=i\right]$$

$$= \mathop{\mathbb{E}}_{\mathbf{h}'\sim P} \sum_{i=0}^{m} \mathop{\mathbb{P}}_{S'\sim\mathcal{D}^m}\left(|S'_{(\mathbf{h},0)}|=i\right) \mathop{\mathbb{E}}_{\widetilde{S}\sim\mathcal{D}_{(\mathbf{h},0)}^i} e^{i\cdot\mathrm{kl}\left(R_{\widetilde{S}}(\mathbf{h}')\middle\|R_{\mathcal{D}_{(\mathbf{h},0)}}(\mathbf{h}')\right)} \quad \langle\text{ Def. of } \mathcal{D}_{(\mathbf{h},0)}\rangle$$

$$\leq \mathop{\mathbb{E}}_{\mathbf{h}'\sim P} \sum_{i=0}^{m} \mathop{\mathbb{P}}_{S'\sim\mathcal{D}^m}\left(|S'_{(\mathbf{h},0)}|=i\right) \sum_{j=0}^{i}\binom{i}{j}\left(R_{\mathcal{D}_{(\mathbf{h},0)}}(\mathbf{h}')\right)^j\left(1-R_{\mathcal{D}_{(\mathbf{h},0)}}(\mathbf{h}')\right)^{i-j} e^{i\cdot\mathrm{kl}\left(\frac{j}{i}\middle\|R_{\mathcal{D}_{(\mathbf{h},0)}}(\mathbf{h}')\right)} \quad \langle\text{Lemma 15}\rangle$$

$$\leq \mathop{\mathbb{E}}_{\mathbf{h}'\sim P} \sum_{i=0}^{m} \mathop{\mathbb{P}}_{S'\sim\mathcal{D}^m}\left(|S'_{(\mathbf{h},0)}|=i\right) \sup_{r\in[0,1]}\sum_{j=0}^{i}\binom{i}{j}(r)^j(1-r)^{i-j} e^{i\cdot\mathrm{kl}\left(\frac{j}{i}\middle\|r\right)}$$

$$\leq \mathop{\mathbb{E}}_{\mathbf{h}'\sim P} \sum_{i=0}^{m} \mathop{\mathbb{P}}_{S'\sim\mathcal{D}^m}\left(|S'_{(\mathbf{h},0)}|=i\right) 2\sqrt{i} \quad \langle\text{Maurer (2004)}\rangle$$

$$\leq \mathop{\mathbb{E}}_{\mathbf{h}'\sim P} 2\sqrt{m}$$

$$= 2\sqrt{m}.$$

Plugging this into our previous result, we obtain

$$\mathop{\mathbb{P}}_{S\sim\mathcal{D}^m}\left(\forall Q \text{ over } \mathcal{H} \; : \; \mathrm{kl}\left(\mathrm{o}_S^Q(\mathbf{h})\middle\|\mathrm{o}_\mathcal{D}^Q(\mathbf{h})\right) \leq \frac{1}{|S_{(\mathbf{h},0)}|}\left[\mathrm{KL}(Q\|P) + \ln\left(\frac{2\sqrt{m}}{\delta}\right)\right]\right) \geq 1-\delta. \qquad \square$$

**Corollary 3** (Triple bound – Single hypothesis). *Let $\mathcal{D}$ be a data distribution, $Q$ a distribution over $\mathcal{H}$, $\mathbf{h} \in \mathcal{H}$ and $\delta_1, \delta_2, \delta_3 \in [0, 1]$. Let $\widetilde{R}_S(Q)$, $\widetilde{\mathfrak{o}}_S^Q(\mathbf{h})$ and $\widetilde{\mathfrak{i}}_S^Q(\mathbf{h})$ be, respectively, probabilistic upper, lower, and lower bounds on $R_{\mathcal{D}}(Q)$, $\mathfrak{o}_{\mathcal{D}}^Q(\mathbf{h})$, and $\mathfrak{i}_{\mathcal{D}}^Q(\mathbf{h})$ such that*

$$(1) \quad \mathbb{P}_{S \sim \mathcal{D}^m} \left( \forall Q \text{ over } \mathcal{H} : R_{\mathcal{D}}(Q) \leq \widetilde{R}_S(Q) \right) \geq 1 - \delta_1,$$

$$(2) \quad \mathbb{P}_{S \sim \mathcal{D}^m} \left( \forall Q \text{ over } \mathcal{H} \ : \ \mathfrak{o}_{\mathcal{D}}^Q(\mathbf{h}) \geq \widetilde{\mathfrak{o}}_S^Q(\mathbf{h}) \right) \geq 1 - \delta_2,$$

$$(3) \quad \mathbb{P}_{S \sim \mathcal{D}^m} \left( \forall Q \text{ over } \mathcal{H} \ : \ \mathfrak{i}_{\mathcal{D}}^Q(\mathbf{h}) \geq \widetilde{\mathfrak{i}}_S^Q(\mathbf{h}) \right) \geq 1 - \delta_3,$$

$$(4) \quad \widetilde{\mathfrak{i}}_S^Q(\mathbf{h}) > \widetilde{\mathfrak{o}}_S^Q(\mathbf{h}).$$

*Then,*

$$\mathbb{P}_{S \sim \mathcal{D}^m} \left( \forall Q \text{ over } \mathcal{H} \ : \ R_{\mathcal{D}}(\mathbf{h}) \leq \frac{\widetilde{R}_S(Q) - \widetilde{\mathfrak{o}}_S^Q(\mathbf{h})}{\widetilde{\mathfrak{i}}_S^Q(\mathbf{h}) - \widetilde{\mathfrak{o}}_S^Q(\mathbf{h})} \right) \geq 1 - \delta,$$

*where $\delta = \delta_1 + \delta_2 + \delta_3$.*

*Proof.* Given the assumptions and knowing that either $\mathfrak{o}_{\mathcal{D}}^Q(\mathbf{h}) \leq R_{\mathcal{D}}(Q) \leq \mathfrak{i}_{\mathcal{D}}^Q(\mathbf{h})$ or $\mathfrak{i}_{\mathcal{D}}^Q(\mathbf{h}) \leq R_{\mathcal{D}}(Q) \leq \mathfrak{o}_{\mathcal{D}}^Q(\mathbf{h})$ is necessary to ensure $R_{\mathcal{D}}(\mathbf{h}) \in [0, 1]$, we obtain from Proposition 1 and the union bound, with probability at least $1 - \delta$:

$$R_{\mathcal{D}}(\mathbf{h}) \ = \ \frac{R_{\mathcal{D}}(Q) - \mathfrak{o}_{\mathcal{D}}^Q(\mathbf{h})}{\mathfrak{i}_{\mathcal{D}}^Q(\mathbf{h}) - \mathfrak{o}_{\mathcal{D}}^Q(\mathbf{h})} \ \leq \ \frac{R_{\mathcal{D}}(Q) - \widetilde{\mathfrak{o}}_S^Q(\mathbf{h})}{\mathfrak{i}_{\mathcal{D}}^Q(\mathbf{h}) - \widetilde{\mathfrak{o}}_S^Q(\mathbf{h})} \ \leq \ \frac{\widetilde{R}_S(Q) - \widetilde{\mathfrak{o}}_S^Q(\mathbf{h})}{\widetilde{\mathfrak{i}}_S^Q(\mathbf{h}) - \widetilde{\mathfrak{o}}_S^Q(\mathbf{h})}.$$

$\square$

**Corollary 4** (Triple bound). *Let $\mathcal{D}$ be a data distribution, $Q$ a distribution over $\mathcal{H}$ and $\delta \in [0, 1]$. Let $\widetilde{R}_S(Q)$, $\widetilde{\mathfrak{o}}_S^Q(\mathbf{h})$ and $\widetilde{\mathfrak{i}}_S^Q(\mathbf{h})$ be such that*

$$(1) \quad \mathbb{P}_{S \sim \mathcal{D}^m} \left( \forall Q \text{ over } \mathcal{H} \ : \ R_{\mathcal{D}}(Q) \leq \widetilde{R}_S(Q) \right) \geq 1 - \delta,$$

$$(2) \quad \forall Q \text{ over } \mathcal{H}, \mathbf{h} \in \mathcal{H} \ : \ \mathfrak{o}_{\mathcal{D}}^Q(\mathbf{h}) \geq \widetilde{\mathfrak{o}}_S^Q(\mathbf{h}) \text{ and } \mathfrak{i}_{\mathcal{D}}^Q(\mathbf{h}) \geq \widetilde{\mathfrak{i}}_S^Q(\mathbf{h}),$$

$$(3) \quad \widetilde{\mathfrak{i}}_S^Q(\mathbf{h}) > \widetilde{\mathfrak{o}}_S^Q(\mathbf{h}).$$

*Then,*

$$\mathbb{P}_{S \sim \mathcal{D}^m} \left( \forall Q \text{ over } \mathcal{H}, \mathbf{h} \in \mathcal{H} \ : \ R_{\mathcal{D}}(\mathbf{h}) \leq \frac{\widetilde{R}_S(Q) - \widetilde{\mathfrak{o}}_S^Q(\mathbf{h})}{\widetilde{\mathfrak{i}}_S^Q(\mathbf{h}) - \widetilde{\mathfrak{o}}_S^Q(\mathbf{h})} \right) \geq 1 - \delta.$$

*Proof.* (1) can be rewritten as

$$\mathbb{P}_{S \sim \mathcal{D}^m} \left( \forall Q \text{ over } \mathcal{H}, \mathbf{h} \in \mathcal{H} \ : \ R_{\mathcal{D}}(Q) \leq \widetilde{R}_S(Q) \right) \geq 1 - \delta,$$

since the predicate is independent of $\mathbf{h}$. The main result follows the same steps as Corollary 3's proof. $\square$

**Proposition 5** (Majority vote–Categorical derandomization). *In the context of Proposition 1: let $Q = \mathcal{C}(\mathbf{p})$ be a Categorical distribution with parameters $\mathbf{p}$. For any data distribution $\mathcal{D}$ and $\mathbf{p} \in \{\mathbf{p}' \in [0, 1]^n \mid \sum_{i=1}^n p_i' = 1\}$, we have*

$$\mathbb{E}_{\mathbf{w} \sim \mathcal{C}(\mathbf{p})} \ell(\mathbf{h_w}(\mathbf{x}), \mathbf{y}) = p_{\mathcal{F}},$$

$$\mathfrak{o}_{\mathcal{D}}^{\mathcal{C}(\mathbf{p})}(\mathbf{h_p}) = \mathbb{E}_{(\mathbf{x}, \mathbf{y}) \sim \mathcal{D}} [p_{\mathcal{F}} \mid p_{\mathcal{F}} < 0.5],$$

$$\mathfrak{i}_{\mathcal{D}}^{\mathcal{C}(\mathbf{p})}(\mathbf{h_p}) = \mathbb{E}_{(\mathbf{x}, \mathbf{y}) \sim \mathcal{D}} [p_{\mathcal{F}} \mid p_{\mathcal{F}} \geq 0.5],$$

*where $p_{\mathcal{F}} = \sum_{i=1}^n p_i \mathbb{1}\{\mathbf{f}_i(\mathbf{x}) \neq \mathbf{y}\}$.*

*Proof.* We simplify the inner content of $\mathfrak{o}_{\mathcal{D}}^{Q}(\mathbf{h})$ and $\mathfrak{1}_{\mathcal{D}}^{Q}(\mathbf{h})$ as such:

$$
\begin{aligned}
\mathop{\mathbb{E}}_{\mathbf{w}\sim\mathcal{C}(\mathbf{p})} \ell(\mathbf{h_w}(\mathbf{x}), \mathbf{y}) &= \mathop{\mathbb{E}}_{\mathbf{w}\sim\mathcal{C}(\mathbf{p})} \mathbb{1}\left\{ \sum_{j=1}^{n} w_j \mathbb{1}\left\{ \mathbf{f}_j(\mathbf{x}) \neq \mathbf{y}\right\} \geq 0.5 \right\} \\
&= \sum_{i=1}^{n} p_i \mathbb{1}\{\mathbb{1}\left\{\mathbf{f}_i(\mathbf{x}) \neq \mathbf{y}\right\} \geq 0.5\} \\
&= \sum_{i=1}^{n} p_i \mathbb{1}\{\mathbf{f}_i(\mathbf{x}) \neq \mathbf{y}\} \\
&= p_{\mathcal{F}}
\end{aligned}
$$

By choosing $\mathbf{h} := \mathbf{h_p}$ and given that

$$
\ell(\mathbf{h_p}(\mathbf{x}), \mathbf{y}) = 0 \iff \sum_{i=1}^{n} p_i \mathbb{1}\left\{\mathbf{f}_i(\mathbf{x}) \neq \mathbf{y}\right\} \geq 0.5 \iff p_{\mathcal{F}} \geq 0.5,
$$

$$
\ell(\mathbf{h_p}(\mathbf{x}), \mathbf{y}) = 1 \iff \sum_{i=1}^{n} p_i \mathbb{1}\left\{\mathbf{f}_i(\mathbf{x}) \neq \mathbf{y}\right\} < 0.5 \iff p_{\mathcal{F}} < 0.5,
$$

we obtain the desired results. $\qquad\square$

**Proposition 6** (Weights partitioning lower bound–Categorical)**.** *In the context of Proposition 5: let $\mathbf{p}_1$ and $\mathbf{p}_2$ be the result of the partition problem applied to $\mathbf{p}$. Then,*

$$
\mathfrak{1}_{\mathcal{D}}^{\mathcal{C}(\mathbf{p})}(\mathbf{h_p}) \geq \max\left( \sum_{p\in\mathbf{p}_1} p, \sum_{p\in\mathbf{p}_2} p \right).
$$

*Proof.* Recall that $p_{\mathcal{F}} = \sum_{i=1}^{n} p_i \mathbb{1}\left\{\mathbf{f}_i(\mathbf{x}) \neq \mathbf{y}\right\}$. Thus, for any $(\mathbf{x}, \mathbf{y}) \in \mathcal{X} \times \mathcal{Y}$, there exists $\mathbf{i} \in \{0, 1\}^{n}$ such that $p_{\mathcal{F}} = \mathbf{i} \cdot \mathbf{p}$.

$$
\begin{aligned}
\mathfrak{1}_{\mathcal{D}}^{\mathcal{C}(\mathbf{p})}(\mathbf{h_p}) &= \mathop{\mathbb{E}}_{(\mathbf{x}, \mathbf{y})\sim\mathcal{D}} \left[ p_{\mathcal{F}} \mid p_{\mathcal{F}} \geq 0.5 \right] \\
&\geq \min_{(\mathbf{x}, \mathbf{y})\in\mathcal{X}\times\mathcal{Y}} \left[ p_{\mathcal{F}} \mid p_{\mathcal{F}} \geq 0.5 \right] \\
&\geq \min_{\mathbf{i}\in\{0,1\}^{n}} \left[ \mathbf{i} \cdot \mathbf{p} \mid \mathbf{i} \cdot \mathbf{p} \geq 0.5 \right] \\
&= \max\left( \sum_{p\in\mathbf{p}_1} p, \sum_{p\in\mathbf{p}_2} p \right).
\end{aligned}
$$

The last equality comes from the fact that $\min_{\mathbf{i}\in\{0,1\}^{n}} \left[ \mathbf{i} \cdot \mathbf{p} \mid \mathbf{i} \cdot \mathbf{p} \geq 0.5 \right]$ is a reformulation of the partition problem (Mertens, 2005). $\qquad\square$

**Proposition 7** (Majority vote–Dirichlet derandomization)**.** *In the context of Proposition 1: let $Q = D(\mathbf{p})$ be a Dirichlet distribution with parameters $\mathbf{p}$. For any data distribution $\mathcal{D}$ and $\mathbf{p} \in \mathbb{R}_{>0}^{n}$:*

$$
\mathop{\mathbb{E}}_{\mathbf{w}\sim D(\mathbf{p})} \ell(\mathbf{h_w}(\mathbf{x}), \mathbf{y}) = I_{0.5}\big( \|\mathbf{p}\|_1 - p_{\mathcal{F}},\, p_{\mathcal{F}} \big),
$$

$$
\mathfrak{o}_{\mathcal{D}}^{D(\mathbf{p})}(\mathbf{h_p}) = \mathop{\mathbb{E}}_{(\mathbf{x}, \mathbf{y})\sim\mathcal{D}} \left[ I_{0.5}\big( \|\mathbf{p}\|_1 - p_{\mathcal{F}},\, p_{\mathcal{F}} \big) \,\Big|\, p_{\mathcal{F}} < \frac{\|\mathbf{p}\|_1}{2} \right],
$$

$$
\mathfrak{1}_{\mathcal{D}}^{D(\mathbf{p})}(\mathbf{h_p}) = \mathop{\mathbb{E}}_{(\mathbf{x}, \mathbf{y})\sim\mathcal{D}} \left[ I_{0.5}\big( \|\mathbf{p}\|_1 - p_{\mathcal{F}},\, p_{\mathcal{F}} \big) \,\Big|\, p_{\mathcal{F}} \geq \frac{\|\mathbf{p}\|_1}{2} \right],
$$

*where $p_{\mathcal{F}} = \sum_{i=1}^{n} p_i \mathbb{1}\{\mathbf{f}_i(\mathbf{x}) \neq \mathbf{y}\}$ and $I_x(\cdot, \cdot)$ is the regularized incomplete beta function evaluated at $x$.*

*Proof.* We simplify the inner content of $\mathfrak{o}_{\mathcal{D}}^{Q}(\mathbf{h})$ and $\mathfrak{1}_{\mathcal{D}}^{Q}(\mathbf{h})$ as such:

$$\mathop{\mathbb{E}}_{\mathbf{w}\sim D(\mathbf{p})}\ell(\mathbf{h}_{\mathbf{w}}(\mathbf{x}),\mathbf{y}) = \mathop{\mathbb{E}}_{\mathbf{w}\sim D(\mathbf{p})}\mathbb{1}\left\{\sum_{j=1}^{n}w_j\mathbb{1}\left\{\mathbf{f}_j(\mathbf{x})\neq\mathbf{y}\right\}\geq\frac{||\mathbf{p}||_1}{2}\right\}$$

$$= I_{0.5}\left(||\mathbf{p}||_1 - p_{\mathcal{F}}, p_{\mathcal{F}}\right). \qquad\qquad \langle\text{See Zantedeschi et al. (2021)}\rangle$$

By choosing $\mathbf{h}:=\mathbf{h_p}$ and given that

$$\ell(\mathbf{h_p}(\mathbf{x}),\mathbf{y})=0 \;\Leftrightarrow\; \sum_{i=1}^{n}p_i\mathbb{1}\left\{\mathbf{f}_i(\mathbf{x})\neq\mathbf{y}\right\}\geq\frac{||\mathbf{p}||_1}{2} \;\Leftrightarrow\; p_{\mathcal{F}}\geq\frac{||\mathbf{p}||_1}{2},$$

$$\ell(\mathbf{h_p}(\mathbf{x}),\mathbf{y})=1 \;\Leftrightarrow\; \sum_{i=1}^{n}p_i\mathbb{1}\left\{\mathbf{f}_i(\mathbf{x})\neq\mathbf{y}\right\}<\frac{||\mathbf{p}||_1}{2} \;\Leftrightarrow\; p_{\mathcal{F}}<\frac{||\mathbf{p}||_1}{2},$$

we obtain the desired results. $\qquad\qquad\square$

**Proposition 8** (Weights partitioning lower bound–Dirichlet). *In the context of Proposition 7: let $\mathbf{p}_1$ and $\mathbf{p}_2$ be the result of the partition problem applied to $\mathbf{p}$. Let $\widetilde{\mathbf{p}}=\max\left(\sum_{p\in\mathbf{p}_1}p,\sum_{p\in\mathbf{p}_2}p\right)$. Then,*

$$\mathfrak{1}_{\mathcal{D}}^{D(\mathbf{p})}(\mathbf{h_p})\geq I_{0.5}\left(||\mathbf{p}||_1-\widetilde{\mathbf{p}},\;\widetilde{\mathbf{p}}\right).$$

*Proof.* We recall that $p_{\mathcal{F}}=\sum_{i=1}^{n}p_i\mathbb{1}\left\{\mathbf{f}_i(\mathbf{x})\neq\mathbf{y}\right\}$ and thus that for any $(\mathbf{x},\mathbf{y})\in\mathcal{X}\times\mathcal{Y}$, there exists $\mathbf{i}\in\{0,1\}^n$ such that $p_{\mathcal{F}}=\mathbf{i}\cdot\mathbf{p}$.

$$\mathfrak{1}_{\mathcal{D}}^{D(\mathbf{p})}(\mathbf{h_p}) = \mathop{\mathbb{E}}_{(\mathbf{x},\mathbf{y})\sim\mathcal{D}}\left[I_{0.5}\left(||\mathbf{p}||_1-p_{\mathcal{F}},p_{\mathcal{F}}\right)\Big|p_{\mathcal{F}}\geq\frac{||\mathbf{p}||_1}{2}\right]$$

$$\geq \min_{(\mathbf{x},\mathbf{y})\in\mathcal{X}\times\mathcal{Y}}\left[I_{0.5}\left(||\mathbf{p}||_1-p_{\mathcal{F}},p_{\mathcal{F}}\right)\Big|p_{\mathcal{F}}\geq\frac{||\mathbf{p}||_1}{2}\right]$$

$$\geq \min_{\mathbf{i}\in\{0,1\}^n}\left[I_{0.5}\left(||\mathbf{p}||_1-\mathbf{i}\cdot\mathbf{p},\mathbf{i}\cdot\mathbf{p}\right)\Big|\mathbf{i}\cdot\mathbf{p}\geq\frac{||\mathbf{p}||_1}{2}\right].$$

Since $I_{0.5}(\cdot,\cdot)$ is decreasing in its first argument, and increasing in its second one, we have

$$\operatorname*{argmin}_{\mathbf{i}\in\{0,1\}^n}\left[I_{0.5}\left(||\mathbf{p}||_1-\mathbf{i}\cdot\mathbf{p},\mathbf{i}\cdot\mathbf{p}\right)\Big|\mathbf{i}\cdot\mathbf{p}\geq\frac{||\mathbf{p}||_1}{2}\right] = \min_{\mathbf{i}\in\{0,1\}^n}\left[\mathbf{i}\cdot\mathbf{p}\;\Big|\;\mathbf{i}\cdot\mathbf{p}\geq\frac{||\mathbf{p}||_1}{2}\right] = \max\left(\sum_{p\in\mathbf{p}_1}p,\sum_{p\in\mathbf{p}_2}p\right)=\widetilde{\mathbf{p}}.$$

Thus, substituting in our first development:

$$\mathfrak{1}_{\mathcal{D}}^{D(\mathbf{p})}(\mathbf{h_p})\geq \min_{\mathbf{i}\in\{0,1\}^n}\left[I_{0.5}\left(||\mathbf{p}||_1-\mathbf{i}\cdot\mathbf{p},\mathbf{i}\cdot\mathbf{p}\right)\Big|\mathbf{i}\cdot\mathbf{p}\geq\frac{||\mathbf{p}||_1}{2}\right]$$

$$= I_{0.5}\left(||\mathbf{p}||_1-\widetilde{\mathbf{p}},\widetilde{\mathbf{p}}\right).$$

$\qquad\qquad\square$

**Proposition 9** (Majority vote–Binary Gaussian derandomization). *In the context of Proposition 1: let $Q=\mathcal{N}(\mathbf{p},\mathrm{I})$ be a Gaussian distribution with mean $\mathbf{p}$ and identity covariance matrix. For any data distribution $\mathcal{D}$ and $\mathbf{p}\in\mathbb{R}^n$, we have*

$$\mathop{\mathbb{E}}_{\mathbf{w}\sim\mathcal{N}(\mathbf{p},\mathrm{I})}\ell(\mathbf{h}_{\mathbf{w}}(\mathbf{x}),y) = \Phi\left(y\frac{\mathbf{p}\cdot\mathbf{f}(\mathbf{x})}{||\mathbf{f}(\mathbf{x})||}\right),$$

$$\mathfrak{1}_{\mathcal{D}}^{\mathcal{N}(\mathbf{p},\mathrm{I})}(\mathbf{h_p}) = 1-\mathop{\mathbb{E}}_{(\mathbf{x},y)\sim\mathcal{D}}\left[\Phi\left(\frac{|\mathbf{p}\cdot\mathbf{f}(\mathbf{x})|}{||\mathbf{f}(\mathbf{x})||}\right)\Big|y(\mathbf{p}\cdot\mathbf{f}(\mathbf{x}))\leq 0\right],$$

$$\mathfrak{o}_{\mathcal{D}}^{\mathcal{N}(\mathbf{p},\mathrm{I})}(\mathbf{h_p}) = \mathop{\mathbb{E}}_{(\mathbf{x},y)\sim\mathcal{D}}\left[\Phi\left(\frac{|\mathbf{p}\cdot\mathbf{f}(\mathbf{x})|}{||\mathbf{f}(\mathbf{x})||}\right)\Big|y(\mathbf{p}\cdot\mathbf{f}(\mathbf{x}))> 0\right],$$

with $\Phi(k) = \frac{1}{2}\left(1 - \text{erf}\left(\frac{k}{\sqrt{2}}\right)\right)$, $\text{erf}(k) = \frac{2}{\sqrt{\pi}}\int_0^k e^{-t^2}dt$.

*Proof.* We simplify the inner content of $\mathfrak{o}_{\mathcal{D}}^Q(h)$ and $\mathfrak{1}_{\mathcal{D}}^Q(h)$ as such:

$$
\mathbb{E}_{\mathbf{w}\sim\mathcal{N}(\mathbf{p},\mathbf{I})} \ell(\mathbf{h_w}(\mathbf{x}), y) = \mathbb{E}_{\mathbf{w}\sim\mathcal{N}(\mathbf{p},\mathbf{I})} \mathbb{1}\left\{ y \neq \underset{\hat{y}\in\mathcal{Y}}{\text{argmax}} \sum_{j=1}^n w_j \mathbb{1}\{f_j(\mathbf{x}) = \hat{y}\} \right\}
$$

$$
= \mathbb{E}_{\mathbf{w}\sim\mathcal{N}(\mathbf{p},\mathbf{I})} \frac{1}{2}\left(1 - y\,\text{sgn}(\mathbf{w}^\top \mathbf{f}(\mathbf{x}))\right)
$$

$$
= \Phi\left(y\frac{\mathbf{p}\cdot\mathbf{f}(\mathbf{x})}{\|\mathbf{f}(\mathbf{x})\|}\right). \qquad\qquad \langle\text{See Langford \& Shawe-Taylor (2002)}\rangle
$$

By choosing $h := \mathbf{h_p}$:

$$
\mathfrak{1}_{\mathcal{D}}^{\mathcal{N}(\mathbf{p},\mathbf{I})}(\mathbf{h_p}) = \mathbb{E}_{(\mathbf{x},y)\sim\mathcal{D}}\left[ \mathbb{E}_{\mathbf{w}\sim\mathcal{N}(\mathbf{p},\mathbf{I})} \ell(\mathbf{h_w}(\mathbf{x}), y)\ \Big|\ \ell(\mathbf{h_p}(\mathbf{x}), y) = 1 \right]
$$

$$
= \mathbb{E}_{(\mathbf{x},y)\sim\mathcal{D}}\left[ \Phi\left(y\frac{\mathbf{p}\cdot\mathbf{f}(\mathbf{x})}{\|\mathbf{f}(\mathbf{x})\|}\right)\ \Big|\ \frac{1}{2}\left(1 - y\,\text{sgn}(\mathbf{p}\cdot\mathbf{f}(\mathbf{x}))\right) = 1 \right]
$$

$$
= \mathbb{E}_{(\mathbf{x},y)\sim\mathcal{D}}\left[ \Phi\left(\frac{y\,\mathbf{p}\cdot\mathbf{f}(\mathbf{x})}{\|\mathbf{f}(\mathbf{x})\|}\right)\ \Big|\ y\,\mathbf{p}\cdot\mathbf{f}(\mathbf{x}) \leq 0 \right]
$$

$$
= \mathbb{E}_{(\mathbf{x},y)\sim\mathcal{D}}\left[ \Phi\left(\frac{-|y\,\mathbf{p}\cdot\mathbf{f}(\mathbf{x})|}{\|\mathbf{f}(\mathbf{x})\|}\right)\ \Big|\ y\,\mathbf{p}\cdot\mathbf{f}(\mathbf{x}) \leq 0 \right]
$$

$$
= \mathbb{E}_{(\mathbf{x},y)\sim\mathcal{D}}\left[ 1 - \Phi\left(\frac{|\mathbf{p}\cdot\mathbf{f}(\mathbf{x})|}{\|\mathbf{f}(\mathbf{x})\|}\right)\ \Big|\ y\,\mathbf{p}\cdot\mathbf{f}(\mathbf{x}) \leq 0 \right]
$$

$$
= 1 - \mathbb{E}_{(\mathbf{x},y)\sim\mathcal{D}}\left[ \Phi\left(\frac{|\mathbf{p}\cdot\mathbf{f}(\mathbf{x})|}{\|\mathbf{f}(\mathbf{x})\|}\right)\ \Big|\ y\,\mathbf{p}\cdot\mathbf{f}(\mathbf{x}) \leq 0 \right].
$$

$$
\mathfrak{o}_{\mathcal{D}}^{\mathcal{N}(\mathbf{p},\mathbf{I})}(\mathbf{h_p}) = \mathbb{E}_{(\mathbf{x},y)\sim\mathcal{D}}\left[ \mathbb{E}_{\mathbf{w}\sim\mathcal{N}(\mathbf{p},\mathbf{I})} \ell(\mathbf{h_w}(\mathbf{x}), y)\ \Big|\ \ell(\mathbf{h_p}(\mathbf{x}), y) = 0 \right]
$$

$$
= \mathbb{E}_{(\mathbf{x},y)\sim\mathcal{D}}\left[ \Phi\left(y\frac{\mathbf{p}\cdot\mathbf{f}(\mathbf{x})}{\|\mathbf{f}(\mathbf{x})\|}\right)\ \Big|\ \frac{1}{2}\left(1 - y\,\text{sgn}(\mathbf{p}\cdot\mathbf{f}(\mathbf{x}))\right) = 0 \right]
$$

$$
= \mathbb{E}_{(\mathbf{x},y)\sim\mathcal{D}}\left[ \Phi\left(\frac{y\,\mathbf{p}\cdot\mathbf{f}(\mathbf{x})}{\|\mathbf{f}(\mathbf{x})\|}\right)\ \Big|\ y\,\mathbf{p}\cdot\mathbf{f}(\mathbf{x}) > 0 \right]
$$

$$
= \mathbb{E}_{(\mathbf{x},y)\sim\mathcal{D}}\left[ \Phi\left(\frac{|y\,\mathbf{p}\cdot\mathbf{f}(\mathbf{x})|}{\|\mathbf{f}(\mathbf{x})\|}\right)\ \Big|\ y\,\mathbf{p}\cdot\mathbf{f}(\mathbf{x}) > 0 \right]
$$

$$
= \mathbb{E}_{(\mathbf{x},y)\sim\mathcal{D}}\left[ \Phi\left(\frac{|\mathbf{p}\cdot\mathbf{f}(\mathbf{x})|}{\|\mathbf{f}(\mathbf{x})\|}\right)\ \Big|\ y\,\mathbf{p}\cdot\mathbf{f}(\mathbf{x}) > 0 \right].
$$

$\qquad\qquad\qquad\qquad\qquad\qquad\qquad\qquad\qquad\qquad\qquad\qquad\qquad\qquad\qquad\qquad\qquad$ □

**Proposition 10** (Weights partitioning lower bound–Binary Gaussian)**.** *In the context of Proposition 9: let* $\mathbf{p}_1$ *and* $\mathbf{p}_2$ *be the result of the partition problem applied to* $\mathbf{p}$. *Let* $\bar{p} = \left|\sum_{p\in\mathbf{p}_1} p - \sum_{p\in\mathbf{p}_2} p\right|$. *Then,*

$$
\mathfrak{1}_{\mathcal{D}}^{\mathcal{N}(\mathbf{p},\mathbf{I})}(\mathbf{h_p}) \geq 1 - \Phi\left(\frac{\bar{p}}{\sqrt{n}}\right).
$$

*Proof.* Since every base classifier has a prediction in $\{-1, +1\}$, for every $\mathbf{x}$, we have $||\mathbf{f}(\mathbf{x})|| = \sqrt{n}$. We have:

$$
\begin{aligned}
\mathtt{1}_{\mathcal{D}}^{\mathcal{N}(\mathbf{p}, \mathrm{I})}(\mathbf{h}_{\mathbf{p}}) &= 1 - \mathop{\mathbb{E}}_{(\mathbf{x}, y) \sim \mathcal{D}} \left[ \Phi \left( \frac{|\mathbf{p} \cdot \mathbf{f}(\mathbf{x})|}{\sqrt{n}} \right) \;\Big|\; y\, \mathbf{p} \cdot \mathbf{f}(\mathbf{x}) \leq 0 \right] \\
&\geq 1 - \max_{\mathbf{x} \in \mathcal{X}} \Phi \left( \frac{|\mathbf{p} \cdot \mathbf{f}(\mathbf{x})|}{\sqrt{n}} \right) \\
&\geq 1 - \Phi \left( \frac{\min\limits_{\mathbf{x} \in \mathcal{X}} |\mathbf{p} \cdot \mathbf{f}(\mathbf{x})|}{\sqrt{n}} \right) & \langle\ \Phi \text{ is strictly decreasing.} \ \rangle \\
&\geq 1 - \Phi \left( \frac{\min\limits_{\mathbf{i} \in \{-1,1\}^n} |\mathbf{p} \cdot \mathbf{i}|}{\sqrt{n}} \right).
\end{aligned}
$$

Finally, note that

$$
\min_{\mathbf{i} \in \{-1,1\}^n} |\mathbf{p} \cdot \mathbf{i}| = \min_{\mathbf{p}_1, \mathbf{p}_2} \left\{ \Big| \sum_{p \in \mathbf{p}_1} p - \sum_{p \in \mathbf{p}_2} p \Big| : \{\mathbf{p}_1, \mathbf{p}_2\} \text{ is a partition of } \mathbf{p} \right\},
$$

which corresponds to the objective of the partition problem to be minimized. Plugging that into the development yields the main result. □

**Proposition 11** (Weights maximizing lower bound–Binary Gaussian)**.** *In the context of Proposition 9, we have*

$$
\mathtt{o}_{\mathcal{D}}^{\mathcal{N}(\mathbf{p}, \mathrm{I})}(\mathbf{h}_{\mathbf{p}}) \geq \Phi \left( \frac{||\mathbf{p}||_1}{\sqrt{n}} \right).
$$

*Proof.* Since every base classifier has a prediction in $\{-1, +1\}$, for every $\mathbf{x}$, we have $||\mathbf{f}(\mathbf{x})|| = \sqrt{n}$ and $\max_{\mathbf{x} \in \mathcal{X}} |\mathbf{p} \cdot \mathbf{f}(\mathbf{x})| = ||\mathbf{p}||_1$. Thus:

$$
\begin{aligned}
\mathtt{o}_{\mathcal{D}}^{\mathcal{N}(\mathbf{p}, \mathrm{I})}(\mathbf{h}_{\mathbf{p}}) &= \mathop{\mathbb{E}}_{(\mathbf{x}, y) \sim \mathcal{D}} \left[ \Phi \left( \frac{|\mathbf{p} \cdot \mathbf{f}(\mathbf{x})|}{\sqrt{n}} \right) \;\Big|\; y\, \mathbf{p} \cdot \mathbf{f}(\mathbf{x}) > 0 \right] \\
&\geq \min_{\mathbf{x} \in \mathcal{X}} \Phi \left( \frac{|\mathbf{p} \cdot \mathbf{f}(\mathbf{x})|}{\sqrt{n}} \right) \\
&\geq \Phi \left( \frac{\max\limits_{\mathbf{x} \in \mathcal{X}} |\mathbf{p} \cdot \mathbf{f}(\mathbf{x})|}{\sqrt{n}} \right) & \langle\ \Phi \text{ is strictly decreasing.} \ \rangle \\
&\geq \Phi \left( \frac{||\mathbf{p}||_1}{\sqrt{n}} \right).
\end{aligned}
$$

□

**Lemma 16.** *If $\mathbf{w} \sim \mathcal{N}(\boldsymbol{\mu}, \mathrm{I})$, where $\boldsymbol{\mu} \in \mathbb{R}^n$, and $\mathbf{a}_i \in \mathbb{R}^n$ for $i \in \{1, \ldots, k\}$, then*

$$
(\mathbf{w}^\top \mathbf{a}_1, \ldots, \mathbf{w}^\top \mathbf{a}_k) \sim \mathcal{N}(\widehat{\boldsymbol{\mu}}, \widehat{\Sigma}),
$$

*where $\widehat{\boldsymbol{\mu}} = [\boldsymbol{\mu}^\top \mathbf{a}_1, \ldots, \boldsymbol{\mu}^\top \mathbf{a}_k]$ and $\widehat{\Sigma}_{i,j} = \mathbf{a}_i^\top \mathbf{a}_j$.*

**Proposition 12** (Majority vote–Multivariate Gaussian stochastic risk)**.** *In the context of Proposition 1: let $Q = \mathcal{N}(\mathbf{p}, \mathrm{I})$ be a Gaussian distribution with mean $\mathbf{p}$ and identity covariance matrix. Let $|\mathcal{Y}| = k$. For any data distribution $\mathcal{D}$ and $\mathbf{p} \in \mathbb{R}^n$:*

$$
\mathop{\mathbb{E}}_{\mathbf{w} \sim \mathcal{N}(\mathbf{p}, \mathrm{I})} \ell(\mathbf{h}_{\mathbf{w}}(\mathbf{x}), \mathbf{y}) = \sum_{i=1}^{k} \mathbb{1}\{y_i = 1\} F_{Z_i}(\mathbf{0}),
$$

*where $F$ is the cumulative distribution function, $Z_i \sim \mathcal{N}(\boldsymbol{\mu}_i, \Sigma_i)$ is a $(k-1)$-variate Gaussian distribution with*

$$
\mu_{i,j} = \begin{cases} \mathbf{p} \cdot (\mathbf{f}_{:,j}(\mathbf{x}) - \mathbf{f}_{:,i}(\mathbf{x})) & \textit{if } j \in \{1, \dots, i-1\}, \\ \mathbf{p} \cdot (\mathbf{f}_{:,j+1}(\mathbf{x}) - \mathbf{f}_{:,i}(\mathbf{x})) & \textit{if } j \in \{i, \dots, k-1\}, \end{cases}
$$

$$
\hat{\Sigma}_{i,j,k} = \begin{cases} (\mathbf{f}_{:,j}(\mathbf{x}) - \mathbf{f}_{:,i}(\mathbf{x})) \cdot (\mathbf{f}_{:,k}(\mathbf{x}) - \mathbf{f}_{:,i}(\mathbf{x})) & \textit{if } j < i, k < i, \\ (\mathbf{f}_{:,j+1}(\mathbf{x}) - \mathbf{f}_{:,i}(\mathbf{x})) \cdot (\mathbf{f}_{:,k}(\mathbf{x}) - \mathbf{f}_{:,i}(\mathbf{x})) & \textit{if } j \geq i, k < i, \\ (\mathbf{f}_{:,j}(\mathbf{x}) - \mathbf{f}_{:,i}(\mathbf{x})) \cdot (\mathbf{f}_{:,k+1}(\mathbf{x}) - \mathbf{f}_{:,i}(\mathbf{x})) & \textit{if } j < i, k \geq i, \\ (\mathbf{f}_{:,j+1}(\mathbf{x}) - \mathbf{f}_{:,i}(\mathbf{x})) \cdot (\mathbf{f}_{:,k+1}(\mathbf{x}) - \mathbf{f}_{:,i}(\mathbf{x})) & \textit{if } j \geq i, k \geq i. \end{cases}
$$

*Proof.* First notice that

$$
\mathbb{1}\left\{ \mathbf{y} \neq \underset{\hat{\mathbf{y}} \in \mathcal{Y}}{\operatorname{argmax}} \sum_{j=1}^{n} p_j \mathbb{1}\{\mathbf{f}_j(\mathbf{x}) = \hat{\mathbf{y}}\} \right\} = \mathbf{y} \cdot \operatorname{hmax}(p_1 \mathbf{f}_1(\mathbf{x}) + \dots + p_n \mathbf{f}_n(\mathbf{x})),
$$

where

$$
\forall \mathbf{a} \in \mathbb{R}^n, i \in \{1, \dots, n\} \; : \; \operatorname{hmax}_i(\mathbf{a}) = \begin{cases} 1 & \text{if } a_i > a_j \forall j \neq i, \\ 0 & \text{otherwise}. \end{cases}
$$

In words, hmax computes the hard-max of an entry vector, returning a one-hot vector. Without loss of generality, suppose $\mathbf{y}$ is a one-hot vector with value 1 at position $i$. We denote

$$
\mathbf{f}_{:,i}(\cdot) = [f_{1,i}(\cdot), \dots, f_{n,i}(\cdot)].
$$

Then,

$$
\begin{aligned}
\underset{\mathbf{w} \sim \mathcal{N}(\mathbf{p}, \mathbf{I})}{\mathbb{E}} \ell(\mathbf{h}_{\mathbf{w}}(\mathbf{x}), \mathbf{y}) &= \underset{\mathbf{w} \sim \mathcal{N}(\mathbf{p}, \mathbf{I})}{\mathbb{E}} \mathbb{1}\left\{ \mathbf{y} \neq \underset{\hat{\mathbf{y}} \in \mathcal{Y}}{\operatorname{argmax}} \sum_{j=1}^{n} w_j \mathbb{1}\{\mathbf{f}_j(\mathbf{x}) = \hat{\mathbf{y}}\} \right\} \\
&= \underset{\mathbf{w} \sim \mathcal{N}(\mathbf{p}, \mathbf{I})}{\mathbb{E}} \mathbf{y} \cdot \operatorname{hmax}(w_1 \mathbf{f}_1(\mathbf{x}) + \dots + w_n \mathbf{f}_n(\mathbf{x})) \\
&= \mathbf{y} \cdot \underset{\mathbf{w} \sim \mathcal{N}(\mathbf{p}, \mathbf{I})}{\mathbb{E}} \operatorname{hmax}(w_1 \mathbf{f}_1(\mathbf{x}) + \dots + w_n \mathbf{f}_n(\mathbf{x})) \\
&= \sum_{i=1}^{n} \mathbb{1}\{y_i = 1\} \underset{\mathbf{w} \sim \mathcal{N}(\mathbf{p}, \mathbf{I})}{\mathbb{E}} \operatorname{hmax}_i(w_1 \mathbf{f}_1(\mathbf{x}) + \dots + w_n \mathbf{f}_n(\mathbf{x})) \\
&= \sum_{i=1}^{n} \mathbb{1}\{y_i = 1\} \underset{\mathbf{w} \sim \mathcal{N}(\mathbf{p}, \mathbf{I})}{\mathbb{P}} (\operatorname{hmax}_i(w_1 \mathbf{f}_1(\mathbf{x}) + \dots + w_n \mathbf{f}_n(\mathbf{x})) = 1) \\
&= \sum_{i=1}^{n} \mathbb{1}\{y_i = 1\} \underset{\mathbf{w} \sim \mathcal{N}(\mathbf{p}, \mathbf{I})}{\mathbb{P}} (\operatorname{hmax}_i(\mathbf{w} \cdot \mathbf{f}_{:,1}(\mathbf{x}), \dots, \mathbf{w} \cdot \mathbf{f}_{:,n}(\mathbf{x})) = 1) \\
&= \sum_{i=1}^{n} \mathbb{1}\{y_i = 1\} \underset{\mathbf{w} \sim \mathcal{N}(\mathbf{p}, \mathbf{I})}{\mathbb{P}} \left( \bigwedge_{j \neq i} \mathbf{w} \cdot \mathbf{f}_{:,i}(\mathbf{x}) > \mathbf{w} \cdot \mathbf{f}_{:,j}(\mathbf{x}) \right) \\
&= \sum_{i=1}^{n} \mathbb{1}\{y_i = 1\} \underset{\mathbf{w} \sim \mathcal{N}(\mathbf{p}, \mathbf{I})}{\mathbb{P}} \left( \bigwedge_{j \neq i} \mathbf{w} \cdot (\mathbf{f}_{:,j}(\mathbf{x}) - \mathbf{f}_{:,i}(\mathbf{x})) \leq 0 \right) \\
&= \sum_{i=1}^{n} \mathbb{1}\{y_i = 1\} F_{Z_i}(\mathbf{0}),
\end{aligned}
$$

where $F$ is the cumulative distribution function. The final result is obtained by using Lemma 16 to find the distribution of $Z_i$. $\qquad\square$

Table 4: Details of the datasets used in the experimental section coming from either the UCI datasets repository (Dua & Graff, 2017), the LIBSVM library (Chang & Lin, 2011), or Zalando (Xiao et al., 2017). $d$ is the number of features, $k$ then number of classes and $m$ the total number of examples.

| # | Dataset | Full name | Source | $d$ | $k$ | $m$ |
|---|---------|-----------|--------|-----|-----|-----|
| 1 | FASHION | Fashion-MNIST | Zalando | 784 | 10 | 70 000 |
| 2 | MNIST | MNIST | LIBSVM | 784 | 10 | 70 000 |
| 3 | PENDIG | Pendigits | UCI | 9 | 10 | 12 992 |
| 4 | PROTEIN | Protein | LIBSVM | 357 | 3 | 24 387 |
| 5 | SENSOR | Sensorless | LIBSVM | 48 | 11 | 58 509 |
| 6 | ADULT | Adult | LIBSVM | 123 | 2 | 32 561 |
| 7 | CODRNA | CodRNA | LIBSVM | 8 | 2 | 59 535 |
| 8 | HABER | Haberman | UCI | 3 | 2 | 306 |
| 9 | MUSH | Mushroom | UCI | 22 | 2 | 8124 |
| 10 | PHIS | Phishing | LIBSVM | 68 | 2 | 2456 |
| 11 | SVMG | Svmguide1 | LIBSVM | 4 | 2 | 7089 |
| 12 | TTT | TicTacToe | UCI | 9 | 2 | 958 |

## C  Details about the experimental section

We used an NVIDIA GeForce RTX 2080 Ti graphics card for the experiments. We used a batch size equal to 1024, and a learning rate equal to 0.1 with a scheduler reducing this parameter by a factor of 10 with a 2 epochs patience. The maximal number of epochs is set to 100, and patience is set to 25 for performing early stopping.

Table 4 presents the datasets used in the experiments.

