# OpenReview forum: "A Framework for PAC-Bayes Derandomization with Applications to Majority Votes"
_TMLR — Under review for TMLR_

### Review · Reviewer_KvHN · 2026-06-08

**Summary Of Contributions:**

This paper studies an important limitation of standard PAC Bayes bounds. Classical PAC Bayes bounds usually control the expected risk of a randomized classifier drawn from a posterior distribution. In practice, however, one often wants to deploy a single deterministic classifier. The paper proposes a general framework for converting PAC Bayes guarantees on a randomized predictor into guarantees for one deterministic predictor. The authors call this derandomization. The main technical idea is to relate the deterministic risk of a classifier to the Gibbs risk and to two conditional Gibbs risks, one conditioned on examples where the deterministic classifier is correct and one conditioned on examples where it is wrong. The paper then gives a triple bound that combines a PAC Bayes bound on the Gibbs risk with lower bounds on these two conditional quantities. The framework is then specialized to weighted majority votes under categorical, Dirichlet, and Gaussian posterior distributions. For the majority vote case, the paper derives computable bounds using a number partitioning view. The gap between randomized PAC Bayes certificates and deterministic deployment is real, and the paper addresses it directly. I also found the general identity connecting deterministic risk and Gibbs risk clean and useful. The specialization to majority votes is natural and gives concrete algorithms rather than only abstract theory. The experiments compare the proposed partition bound against several known baselines on binary and multi class datasets, and the reported bounds are often substantially tighter.

The main weakness is that the empirical section does not fully separate the value of the new theory from the value of the post training heuristics used to tighten the bound. The method is also currently limited to majority vote style models, and the strongest generality claim should be stated with more care. Some parts of the experiments need more detail to make the comparison fully convincing, especially the choice of distributions, the training procedure for each baseline, the cost of solving the partition problem, and the effect of the three heuristics. The paper also says the code will be made public only upon acceptance, but for a theory plus empirical paper, reproducibility would be much stronger if code or detailed pseudocode were provided during review.

**Audience:**

Yes

**Audience Explanation:**

Yes. I think this paper would be of interest to a part of the TMLR audience, especially researchers working on PAC Bayes theory, certified generalization, ensemble methods, randomized classifiers, and majority vote learning. The question of how to turn a randomized PAC Bayes guarantee into a guarantee for a deterministic classifier is important and practically relevant. Many deployed systems require deterministic predictions, so a certificate for only the randomized Gibbs classifier can be unsatisfactory.

The paper is not likely to be of broad interest to all machine learning readers, because the results are fairly specialized and the current concrete applications are mainly to majority votes. However, the problem is important within the PAC Bayes community, and the framework may be useful as a building block for future work. The paper also helps clarify the relationship between Gibbs risk, Bayes or majority vote risk, and deterministic certificates, which is useful even beyond the exact algorithms tested here.

**Claims And Evidence:**

Yes

**Claims Explanation:**

The main theoretical claims appear to be supported by the propositions and corollaries. The derivation starts from a simple and clear decomposition of the Gibbs risk, and the later bounds follow by combining this decomposition with PAC Bayes style guarantees and deterministic lower bounds. I did not find an obvious flaw in the high level argument.

The empirical evidence also broadly supports the main claim that the proposed partition bound can be tighter than several existing baselines for deterministic majority votes. In Table 2, the partition bound is reported as the best or tied best bound across the listed datasets, sometimes by a large margin. The experiments also include both binary and multi class tasks and compare against first order, second order, binomial, C bound, and VC dimension based baselines where applicable.

That said, I think some claims are stronger than what the evidence cleanly shows. The paper says the approach “consistently outperforms” baselines and later describes “dominance” across scenarios. This is true for the reported bound values, but not always for test error. For example, on several multi class datasets, the partition bound has a much worse test error than the best baseline, even when the bound is tighter.

The experimental results also depend on post training heuristics that are not fully analyzed. The paper optimizes a Gibbs risk bound during training and then applies clipping, coordinate descent, and rescaling heuristics to improve the final partition bound. Since these heuristics seem important, the authors should show how much each one contributes. Without this, it is hard to tell whether the improvement comes mainly from the new bound or from the heuristic search over weights.

**Requested Changes:**

Clarify the exact scope of the main contribution. The paper presents a general derandomization framework, but the practical computable results are mostly for majority votes and for specific posterior families. The abstract and introduction should make this distinction clearer. I would avoid language that suggests the method is generally practical for arbitrary hypothesis spaces unless the authors give a concrete way to compute the needed lower bounds in those settings.

Add an ablation study for the post training heuristics. The final method uses clipping, coordinate descent, and rescaling after training. Since these are not minor implementation details, the paper should report the bound before heuristics, after each heuristic, and after all heuristics. This would show whether the proposed theory alone gives the gains or whether most of the gains come from the heuristic search.

Give more detail on the partition problem computation. The paper says the partition problem is tractable in the experiments because of the bit length to number of voters ratio. I would like to see actual running times, failure modes if any, and whether exact dynamic programming or heuristics were used in each setting. The paper should also explain how continuous weights are discretized or represented for the partition solver.

Make the comparison to baselines more transparent. The paper reports the best bound attained by a single distribution for each method, but this can hide important choices. Please report which distribution was selected for each baseline and dataset, and whether this selection used only training or validation information. Also explain whether all baselines received comparable optimization effort.

Discuss cases where the bound is tight but the classifier accuracy is poor. In Table 2 and Table 3, some partition bound results have much larger test error than competing methods, especially in multi class settings. This is important because users may care about both a good certificate and a good classifier. The paper should discuss this tradeoff and whether the method is best understood as certificate optimization rather than accuracy optimization.

Clarify the role of data dependent hypothesis construction in the multi class experiments. The random forest voters are trained on splits of the data and then evaluated using a cross bounding scheme. This is reasonable, but the paper should explain more plainly how independence is preserved and how the same data is or is not reused across voter construction, posterior learning, and bound evaluation.

Check the notation and presentation of the conditional PAC Bayes result. The conditional sample sizes (m(h,0)) and (m(h,1)) can be zero. The paper should state what happens in this case. It should also clarify whether Theorem 2 is mainly a conceptual result for fixed (h), while Corollary 4 is the usable result for data dependent (h).

---

### Review · Reviewer_sSDX · 2026-06-14

**Summary Of Contributions:**

The paper introduces a unified framework for converting PAC-Bayesian guarantees into guarantees for an individual hypothesis. Building on this perspective, the authors establish a general oracle bound, from which they derive both a numerical bound and a variant tailored to majority-vote classifiers. The experimental results demonstrate that the proposed framework yields tighter generalization guarantees for single classifiers than several widely used benchmark approaches.

**Additional Comments:**

Practical applicability of Corollary 4 depends on the direct computation of $\tilde{R}_S(Q), \tilde{0}_S^Q(h)$ and  $\tilde{1}_S^Q(h)$. This restricts the applicability of the bounds to a specific set of hypothesis classes.

**Audience:**

Yes

**Audience Explanation:**

The learning theory community will appreciate novel results in PAC-Bayes framework.

**Claims And Evidence:**

No

**Claims Explanation:**

As I understand, the crucial part in the proof of Theorem 2 is the partitioning of the dataset $S$ into $S_{(h, .)}$. It seems that the proof of Theorem 2 assumes that samples in $S_{(h, .)}$ are also i.i.d. The samples in $S$ are drawn i.i.d., and I am trying to understand if the i.i.d. property would still hold after partitioning the samples.

Consider the following toy setup:
1. Let $A$ and $Y$ be i.i.d. Bernoulli(0.5) random variables.
2. Let $B = A \oplus Y$, i.e., $B$ is XOR of $A$ and $Y$. One should note that $B$ is also a Bernoulli(0.5) random variable, and is independent of both $A$ and $Y$.
3. For $i=1,\ldots,m$, we can generate i.i.d. draws of $A, Y$ and $B$. First $m/2$ samples $(x, y)$  are chosen as draws of $(A, Y)$, remaining $m/2$ samples are chosen as draws of $(B, Y)$.
4. Let $\ell(h(x), y) = y$. This means that $S_{h, c} = \\{ (x, y) \in S : y = c \\}$ where $c \in \\{0, 1\\}$.
5. Clearly, conditioned on $Y$, $A$ and $B$ are not conditionally independent. Then, is it justified to assume that $S_{h, c}$ will contain i.i.d. samples?

I may have missed some crucial information here, and I would urge the authors to clarify their position. I will be happy to reassess my evaluation after appropriate clarifications from the authors.

**Requested Changes:**

Minor formatting suggestion: It is better to avoid double brackets while citing references within a bracket. One could use \citet for this.

---

### Review · Reviewer_qvoe · 2026-07-13

**Summary Of Contributions:**

Classical PAC-Bayes bounds usually control the Gibbs risk $R_{\mathcal D}(Q) = \mathbb E_{h\sim Q} R_{\mathcal D}(h)$, where a classifier is sampled from a posterior $Q$. In deployment, however, one often uses a deterministic predictor $h$, such as the majority vote induced by the mean of $Q$. The familiar elementary conversion for a majority vote is $R_{\mathcal D}(h_{\mathrm{MV}}) \leq 2R_{\mathcal D}(Q)$, but this factor-two inequality can be loose.

The paper’s starting point is to condition the stochastic classifier’s error on whether the target deterministic classifier $h$ is correct. It introduces $0_{\mathcal D}^{Q}(h) = \mathbb E[\mathbb E_{h'\sim Q}\ell(h'(x),y) | \ell(h(x),y)=0]$ and $1_{\mathcal D}^{Q}(h) = \mathbb E[\mathbb E_{h'\sim Q}\ell(h'(x),y) | \ell(h(x),y)=1]$. The law of total expectation then gives $R_{\mathcal D}(Q)=0_{\mathcal D}^{Q}(h)(1-R_{\mathcal D}(h)) + 1_{\mathcal D}^{Q}(h)R_{\mathcal D}(h)$, and hence, provided $1_{\mathcal D}^{Q}(h)\neq 0_{\mathcal D}^{Q}(h)$, we have that $R_{\mathcal D}(h) = \frac{R_{\mathcal D}(Q)-0_{\mathcal D}^{Q}(h)}{1_{\mathcal D}^{Q}(h)-0_{\mathcal D}^{Q}(h)}$.

This exact identity is the conceptual heart of the paper. It explains precisely when Gibbs risk is informative about the deterministic classifier: the stochastic classifier must make appreciably more errors on points where $h$ is wrong than on points where $h$ is correct. The classical factor-two result is recovered when $0_{\mathcal D}^{Q}(h)\geq 0$ and $1_{\mathcal D}^{Q}(h)\geq\frac{1}{2}$.

The authors then:
- formulate PAC-Bayes bounds for the two conditional quantities $0_{\mathcal D}^{Q}(h)$ and $1_{\mathcal D}^{Q}(h)$;
- combine an upper bound on Gibbs risk with lower bounds on $0_{\mathcal D}^{Q}(h)$ and $1_{\mathcal D}^{Q}(h)$;
- specialise this construction to majority votes under categorical, Dirichlet and Gaussian distributions;
- derive computable deterministic lower bounds through a balanced-partition problem on the voting weights;
- compare the resulting "partition bound" against first-order, second-order, binomial and C-bound-style baselines.

Main strengths:
1. The central decomposition is simple, exact and illuminating.
2. It unifies several majority-vote constructions.
3. Recovery and refinement of the factor-two bound is well motivated.
4. The paper directly targets certification, not merely accuracy.

Main weaknesses and reservations:
1. The claimed generality is greater than the demonstrated generality. The identity involving $0_{\mathcal D}^{Q}(h)$ and $1_{\mathcal D}^{Q}(h)$ is completely general, but a useful bound requires explicit, nontrivial, and preferably tractable bounds on those two conditional quantities. Outside the structured majority-vote examples, it is not clear how readily such bounds can be obtained.
2. The novelty relative to the PAC-Bayes majority-vote literature needs sharper articulation.
3. The partition bound has an awkward computational and optimisation profile. The balanced-partition problem is combinatorial. Even if it is manageable for the experiments, the manuscript should be clearer about: the computational cost as the number of voters grows, whether the partition quantity can be differentiated through, whether training optimises the eventual deterministic certificate or merely a stochastic surrogate, etc.
4. The empirical evidence appears too restricted for several broad claims. The experiments support the claim that the proposed bound can improve on the selected baselines in the tested settings. They do not, in my view, establish that the proposed approach is generically preferable.
5. The motivation occasionally overstates the inadequacy of classical PAC-Bayes. Existing PAC-Bayesian majority-vote bounds, C-bounds, disintegrated bounds and derandomisation arguments do address deterministic or single-predictor questions, although often with looseness, additional moments, or randomness in the selected hypothesis. The problem is therefore not absence of any deterministic guarantee, but the tightness, form and deployability of existing conversions. A more measured formulation would actually strengthen the paper by making its scientific target more accurate.
6. This work is not a replacement for the C-bound. The two use different descriptions of the error geometry, and it would be mathematically interesting to determine when one description provably dominates the other. This paper mostly compares numerical bounds rather than develops such a structural comparison.

**Audience:**

Yes

**Audience Explanation:**

Although pleae see weakness 2 stated above.

**Broader Impact Concerns:**

None.

**Claims And Evidence:**

No

**Claims Explanation:**

Please see weaknesses 1, 3, and 4 stated above.

**Requested Changes:**

The authors should address weaknesses 1, 2, 3, and 4 stated above.

Also addressing weaknesses 5 and 6 would improve the paper.

In addition the paper would be improved if the authors made a stronger attempt at readability by researchers who are not already experts in PAC-Bayes. A suggestion is to include a small running example, e.g., with a small number of voters with explicit weights.